# APPROXIMATION AND LEARNING WITH DEEP CONVOLUTIONAL MODELS: A KERNEL PERSPECTIVE

**Alberto Bietti**
Center for Data Science, New York University
`alberto.bietti@nyu.edu`

## ABSTRACT

The empirical success of deep convolutional networks on tasks involving high-dimensional data such as images or audio suggests that they can efficiently approximate certain functions that are well-suited for such tasks. In this paper, we study this through the lens of kernel methods, by considering simple hierarchical kernels with two or three convolution and pooling layers, inspired by convolutional kernel networks. These achieve good empirical performance on standard vision datasets, while providing a precise description of their functional space that yields new insights on their inductive bias. We show that the RKHS consists of additive models of interaction terms between patches, and that its norm encourages spatial similarities between these terms through pooling layers. We then provide generalization bounds which illustrate how pooling and patches yield improved sample complexity guarantees when the target function presents such regularities.

## 1 INTRODUCTION

Deep convolutional models have been at the heart of the recent successes of deep learning in problems where the data consists of high-dimensional signals, such as image classification or speech recognition. Convolution and pooling operations have notably contributed to the practical success of these models, yet our theoretical understanding of how they enable efficient learning is still limited.

One key difficulty for understanding such models is the curse of dimensionality: due to the high-dimensionality of the input data, it is hopeless to learn arbitrary functions from samples. For instance, classical non-parametric regression techniques for learning generic target functions typically require either low dimension or very high degrees of smoothness in order to obtain good generalization (*e.g.*, Wainwright, 2019), which makes them impractical for dealing with high-dimensional signals. Thus, further assumptions on the target function are needed to make the problem tractable, and we seek assumptions that make convolutions a useful modeling tool. Various works have studied approximation benefits of depth with models that resemble deep convolutional architectures (Cohen & Shashua, 2017; Mhaskar & Poggio, 2016; Schmidt-Hieber et al., 2020). Nevertheless, while such function classes may provide improved statistical efficiency in theory, it is unclear if there exist efficient algorithms to learn such models, and hence, whether they might correspond to what convolutional networks learn in practice. To overcome this issue, we consider instead function classes based on kernel methods (Schölkopf & Smola, 2001; Wahba, 1990), which are known to be learnable with efficient (polynomial-time) algorithms, such as kernel ridge regression or gradient descent.

We consider "deep" structured kernels known as convolutional kernels, which yield good empirical performance on standard computer vision benchmarks (Li et al., 2019; Mairal, 2016; Mairal et al., 2014; Shankar et al., 2020), and are related to over-parameterized convolutional networks (CNNs) in so-called "kernel regimes" (Arora et al., 2019; Bietti & Mairal, 2019b; Daniely et al., 2016; Garriga-Alonso et al., 2019; Jacot et al., 2018; Novak et al., 2019; Yang, 2019). Such regimes may be seen as providing a first-order description of what common deep models trained with gradient methods may learn. Studying the corresponding function spaces (reproducing kernel Hilbert spaces, or RKHS) may then provide insight into the benefits of various architectural choices. For fully-connected architectures, such kernels are rotation-invariant, and the corresponding RKHSs are well understood in terms of regularity properties on the sphere (Bach, 2017a; Smola et al., 2001), but do not show any major differences between deep and shallow kernels (Bietti & Bach, 2021; Chen & Xu, 2021;

Geifman et al., 2020). In contrast, in this work we show that even in the kernel setting, multiple layers of convolution and pooling operations can be crucial for efficient learning of functions with specific structures that are well-suited for natural signals. Our work paves the way for further studies of the inductive bias of optimization algorithms on deep convolutional networks beyond kernel regimes, for instance by incorporating adaptivity to low-dimensional structure (Bach, 2017a; Chizat & Bach, 2020; Wei et al., 2019) or hierarchical learning (Allen-Zhu & Li, 2020).

We make the following contributions:

- We revisit convolutional kernel networks (Mairal, 2016), finding that simple two or three layers models with Gaussian pooling and polynomial kernels of degree 2-4 at higher layers provide competitive performance with state-of-the-art convolutional kernels such as Myrtle kernels (Shankar et al., 2020) on Cifar10.

- For such kernels, we provide an exact description of the RKHS functions and their norm, illustrating representation benefits of multiple convolutional and pooling layers for capturing additive and interaction models on patches with certain spatial regularities among interaction terms.

- We provide generalization bounds that illustrate the benefits of architectural choices such as pooling and patches for learning additive interaction models with spatial invariance in the interaction terms, namely, improvements in sample complexity by polynomial factors in the size of the input signal.

**Related work.**   Convolutional kernel networks were introduced by Mairal et al. (2014); Mairal (2016). Empirically, they used kernel approximations to improve computational efficiency, while we evaluate the exact kernels in order to assess their best performance, as in (Arora et al., 2019; Li et al., 2019; Shankar et al., 2020). Bietti & Mairal (2019a;b) show invariance and stability properties of its RKHS functions, and provide upper bounds on the RKHS norm for some specific functions (see also Zhang et al., 2017); in contrast, we provide exact characterizations of the RKHS norm, and study generalization benefits of certain architectures. Scetbon & Harchaoui (2020) study statistical properties of simple convolutional kernels without pooling, while we focus on the role of architecture choices with an emphasis on pooling. Cohen & Shashua (2016; 2017); Mhaskar & Poggio (2016); Poggio et al. (2017) study expressivity and approximation with models that resemble CNNs, showing benefits thanks to hierarchy or local interactions, but such models are not known to be learnable with tractable algorithms, while we focus on (tractable) kernels. Regularization properties of convolutional models were also considered in (Gunasekar et al., 2018; Heckel & Soltanolkotabi, 2020), but in different regimes or architectures than ours. Li et al. (2021); Malach & Shalev-Shwartz (2021) study benefits of convolutional networks with efficient algorithms, but do not study the gains of pooling. Du et al. (2018) study sample complexity of learning CNNs, focusing on parametric rather than non-parametric models. Mei et al. (2021) study statistical benefits of global pooling for learning invariant functions, but only consider one layer with full-size patches. Concurrently to our work, Favero et al. (2021); Misiakiewicz & Mei (2021) study benefits of local patches, but focus on one-layer models.

## 2   DEEP CONVOLUTIONAL KERNELS

In this section, we recall the construction of multi-layer convolutional kernels on discrete signals, following most closely the convolutional kernel network (CKN) architectures studied by Mairal (2016); Bietti & Mairal (2019a). These architectures rely crucially on pooling layers, typically with Gaussian filters, which make them empirically effective even with just two convolutional layers. These kernels define function spaces that will be the main focus of our theoretical study of approximation and generalization in the next sections. In particular, when learning a target function of the form $f^*(x) = \sum_i f_i(x)$, we will show that they are able to efficiently exploit two useful properties of $f^*$: (locality) each $f_i$ may depend on only one or a few small localized patches of the signal; (invariance) many different terms $f_i$ may involve the same function applied to different input patches. We provide further background and motivation in Appendix A.

For simplicity, we will focus on discrete 1D input signals, though one may easily extend our results to 2D or higher-dimensional signals. We will assume periodic signals in order to avoid difficulties with border effects, or alternatively, a cyclic domain $\Omega = \mathbb{Z}/|\Omega|\mathbb{Z}$. A convolutional kernel of depth $L$ may then be defined for input signals $x, x' \in L^2(\Omega, \mathbb{R}^p)$ by $K_L(x, x') = \langle \Psi(x), \Psi(x') \rangle$, through the explicit feature map

$$\Psi(x) = A_L M_L P_L \cdots A_1 M_1 P_1 x. \tag{1}$$

Here, $P_\ell, M_\ell$ and $A_\ell$ are linear or non-linear operators corresponding to *patch extraction*, *kernel mapping* and *pooling*, respectively, and are described below. They operate on *feature maps* in $L^2(\Omega_\ell)$ (with $\Omega_0 = \Omega$) with values in different Hilbert spaces, starting from $\mathcal{H}_0 := \mathbb{R}^p$, and are defined below. An illustration of this construction is given in Figure 1.

**Patch extraction.** Given a patch shape $S_\ell \subset \Omega_{\ell-1}$, such as $S_\ell = [-1, 0, 1]$ for one-dimensional patches of size 3, the operator $P_\ell$ is defined for $x \in L^2(\Omega_{\ell-1}, \mathcal{H}_{\ell-1})$ by[1]

$$P_\ell x[u] = (x[u+v])_{v \in S_\ell} \in \mathcal{H}_{\ell-1}^{|S_\ell|}.$$

**Kernel mapping.** The operators $M_\ell$ perform a non-linear embedding of patches into a new Hilbert space using dot-product kernels. We consider homogeneous dot-product kernels given for $z, z' \in \mathcal{H}_{\ell-1}^{|S_\ell|}$ by

$x_\ell = A_\ell M_\ell P_\ell x_{\ell-1}$

downsampling

linear pooling

$M_\ell P_\ell x_{\ell-1}$

dot-product kernel

$x_{\ell-1}$

$P_\ell x_{\ell-1}[u] \in \mathcal{H}_{\ell-1}^{|S_\ell|}$

Figure 1: Convolutional kernel.

$$k_\ell(z, z') = \|z\| \|z'\| \kappa_\ell \left( \frac{\langle z, z' \rangle}{\|z\| \|z'\|} \right) = \langle \varphi_\ell(z), \varphi_\ell(z') \rangle_{\mathcal{H}_\ell},$$
(2)

where $\varphi_\ell : \mathcal{H}_{\ell-1}^{|S_\ell|} \to \mathcal{H}_\ell$ is a feature map for the kernel. The kernel functions take the form $\kappa_\ell(u) = \sum_{j \geq 0} b_j u^j$ with $b_j \geq 0$. This includes the exponential kernel $\kappa(u) = e^{\alpha(u-1)}$ (*i.e.*, Gaussian kernel on the sphere) and the arc-cosine kernel arising from random ReLU features (Cho & Saul, 2009), for which our construction is equivalent to that of the conjugate or NNGP kernel for an infinite-width random ReLU network with the same architecture. The operator $M_\ell$ is then defined pointwise by

$$M_\ell x[u] = \varphi_\ell(x[u]).$$
(3)

At the first layer on image patches, these kernels lead to functional spaces consisting of homogeneous functions with varying degrees of smoothness on the sphere, depending on the properties of the kernel (Bach, 2017a; Smola et al., 2001). At higher layers, our theoretical analysis will also consider simple polynomial kernels such as $k_\ell(z, z') = (\langle z, z' \rangle)^r$, in which case the feature map may be explicitly written in terms of tensor products. For instance, $r = 2$ gives $\varphi_\ell(z) = z \otimes z$ and $\mathcal{H}_\ell = (\mathcal{H}_{\ell-1}^{|S_\ell|})^{\otimes 2} = (\mathcal{H}_{\ell-1} \otimes \mathcal{H}_{\ell-1})^{|S_\ell| \times |S_\ell|}$. See Appendix A.2 for more background on dot-product kernels, their tensor products, and their regularization properties.

**Pooling.** Finally, the pooling operators $A_\ell$ perform local averaging through convolution with a filter $h_\ell[u]$, which we may consider to be symmetric ($h_\ell[-u] =: \bar{h}_\ell[u] = h_\ell[u]$). In practice, the pooling operation is often followed by downsampling by a factor $s_\ell$, in which case the new signal $A_\ell x$ is defined on a new domain $\Omega_\ell$ with $|\Omega_\ell| = |\Omega_{\ell-1}|/s_\ell$, and we may write for $x \in L^2(\Omega_{\ell-1})$ and $u \in \Omega_\ell$,

$$A_\ell x[u] = \sum_{v \in \Omega_{\ell-1}} h_\ell[s_\ell u - v] x[v].$$
(4)

Our experiments consider Gaussian pooling filters with a size and bandwidth proportional to the downsampling factor $s_\ell$, following Mairal (2016), namely, size $2s_\ell + 1$ and bandwidth $\sqrt{2}s_\ell$. In Section 3, we will often assume no downsampling for simplicity, in which case we may see the filter bandwidth as increasing with the layers.

**Links with other convolutional kernels.** We note that our construction closely resembles kernels derived from infinitely wide convolutional networks, known as conjugate or NNGP kernels (Garriga-Alonso et al., 2019; Novak et al., 2019), and is also related to convolutional neural tangent kernels (Arora et al., 2019; Bietti & Mairal, 2019b; Yang, 2019). The Myrtle family of kernels (Shankar et al., 2020) also resembles our models, but they use small average pooling filters instead of Gaussian filters, which leads to deeper architectures due to smaller receptive fields.

---

[1] $L^2(\Omega, \mathcal{H})$ denotes the space of $\mathcal{H}$-valued signals $x$ such that $\|x\|_{L^2(\Omega, \mathcal{H})}^2 := \sum_{u \in \Omega} \|x[u]\|_{\mathcal{H}}^2 < \infty$.

## 3  APPROXIMATION WITH (DEEP) CONVOLUTIONAL KERNELS

In this section, we present our main results on the approximation properties of convolutional kernels, by characterizing functions in the RKHS as well as their norms. We begin with the one-layer case, which does not capture interactions between patches but highlights the role of pooling, before moving multiple layers, where interaction terms play an important role. Proofs are given in Appendix E.

### 3.1  THE ONE-LAYER CASE

We begin by considering the case of a single convolutional layer, which can already help us illustrate the role of patches and pooling. Here, the kernel is given by

$$K_1(x, x') = \langle A\Phi(x), A\Phi(x') \rangle_{L^2(\Omega, \mathcal{H})},$$

with $\Phi(x)[u] = \varphi(x_u)$, where we use the shorthand $x_u = Px[u]$ for the patch at position $u$. We now characterize the RKHS of $K_1$, showing that it consists of additive models of functions in $\mathcal{H}$ defined on patches, with spatial regularities among the terms, induced by the pooling operator $A$. (**Notation**: $A^*$ and $A^\dagger$ denote the adjoint and pseudo-inverse of an operator $A$, respectively.)

**Proposition 1** (RKHS for 1-layer CKN.). *The RKHS of $K_1$ consists of functions $f(x) = \langle G, \Phi(x) \rangle_{L^2(\Omega, \mathcal{H})} = \sum_{u \in \Omega} G[u](x_u)$, with $G \in Range(A^*)$, and with RKHS norm*

$$\|f\|^2_{\mathcal{H}_{K_1}} = \inf_{G \in L^2(\Omega, \mathcal{H})} \|A^{\dagger*}G\|^2_{L^2(\Omega, \mathcal{H})} \quad s.t. \quad f(x) = \sum_{u \in \Omega} G[u](x_u) \tag{5}$$

Note that if $A^*$ is not invertible (for instance in the presence of downsampling), the constraint $G \in \text{Range}(A^*)$ is active and $A^{\dagger*}$ is its pseudo-inverse. In the extreme case of global average pooling, we have $A = (1, \ldots, 1) \otimes Id : L^2(\Omega, \mathcal{H}) \to \mathcal{H}$, so that $G \in \text{Range}(A^*)$ is equivalent to $G[u] = g$ for all $u$, for some fixed $g \in \mathcal{H}$. In this case, the penalty in (5) is simply the squared RKHS norm $\|g\|^2_{\mathcal{H}}$.

In order to understand the norm (5) for general pooling, recall that $A$ is a convolution operator with filter $h_1$, hence its inverse (which we now assume exists for simplicity) may be easily computed in the Fourier basis. In particular, for a patch $z \in \mathbb{R}^{p|S_1|}$, defining the scalar signal $g_z[u] = G[u](z)$, we may write the following using the reproducing property and linearity:

$$A^{\dagger*}G[u](z) = (A^{-1})^\top g_z[u] = \mathcal{F}^{-1} \text{diag}(\mathcal{F}\bar{h}_1)^{-1} \mathcal{F}g_z[u],$$

where $\bar{h}_1[u] := h_1[-u]$ arises from transposition, $\mathcal{F}$ is the discrete Fourier transform, and both $\mathcal{F}$ and $A$ are viewed here as $|\Omega| \times |\Omega|$ matrices. From this expression, we see that by penalizing the RKHS norm of $f$, we are implicitly penalizing the high frequencies of the signals $g_z[u]$ for any $z$, and this regularization is stronger when the pooling filter $h_1$ has a fast spectral decay. For instance, as the spatial bandwidth of $h_1$ increases (approaching a global pooling operation), $\mathcal{F}h_1$ decreases more rapidly, which encourages $g_z[u]$ to be more smooth as a function of $u$, and thus prevents $f$ from relying too much on the location of patches. If instead $h_1$ is very localized in space (*e.g.*, a Dirac filter, which corresponds to no pooling), $g_z[u]$ may vary much more rapidly as a function of $u$, which then allows $f$ to discriminate differently depending on the spatial location. This provides a different perspective on the invariance properties induced by pooling. If we denote $\tilde{G}[u] = A^{\dagger*}G[u]$, the penalty writes

$$\|\tilde{G}\|^2_{L^2(\Omega, \mathcal{H})} = \sum_{u \in \Omega} \|\tilde{G}[u]\|^2_{\mathcal{H}}.$$

Here, the RKHS norm $\|\cdot\|_{\mathcal{H}}$ also controls smoothness, but this time for functions $\tilde{G}[u](\cdot)$ defined on input patches. For homogeneous dot-product kernels of the form (2), the norm takes the form $\|g\|_{\mathcal{H}} = \|T^{-\frac{1}{2}}g\|_{L^2(\mathbb{S}^{d-1})}$, where the regularization operator $T^{-\frac{1}{2}}$ is the self-adjoint inverse square root of the integral operator for the patch kernel restricted to $L^2(\mathbb{S}^{d-1})$. For instance, when the eigenvalues of $T$ decay polynomially, as for arc-cosine kernels, $T^{-\frac{1}{2}}$ behaves like a power of the spherical Laplacian (see Bach, 2017a, and Appendix A.2). Then we may write

$$\|\tilde{G}\|^2_{L^2(\Omega, \mathcal{H})} = \|((A^{-1})^\top \otimes T^{-\frac{1}{2}})G\|^2_{L^2(\Omega) \otimes L^2(\mathbb{S}^{d-1})},$$

which highlights that the norm applies two regularization operators $(A^{-1})^\top$ and $T^{-\frac{1}{2}}$ independently on the spatial variable and the patch variable of $(u, z) \mapsto G[u](z)$, viewed here as an element of $L^2(\Omega) \otimes L^2(\mathbb{S}^{d-1})$.

Table 1: Cifar10 test accuracy with 2-layer convolutional kernels with 3x3 patches and pooling/downsampling sizes [2,5], with different choices of patch kernels $\kappa_1$ and $\kappa_2$. The last model is similar to a 1-layer convolutional kernel. See Section 5 for experimental details.

| $\kappa_1$-$\kappa_2$ | Exp-Exp | Exp-Poly3 | Exp-Poly2 | Poly2-Exp | Poly2-Poly2 | Exp-Lin |
|---|---|---|---|---|---|---|
| Test acc. | 87.9% | 87.7% | 86.9% | 85.1% | 82.2% | 80.9% |

## 3.2 THE MULTI-LAYER CASE

We now study the case of convolutional kernels with more than one convolutional layer. While the patch kernels used at higher layers are typically similar to the ones from the first layer, we show empirically on Cifar10 that they may be replaced by simple polynomial kernels with little loss in accuracy. We then proceed by studying the RKHS of such simplified models, highlighting the role of depth for capturing interactions between different patches via kernel tensor products.

**An empirical study.** Table 1 shows the performance of a given 2-layer convolutional kernel architecture, with different choices of patch kernels $\kappa_1$ and $\kappa_2$. The reference model uses exponential kernels in both layers, following the construction in Mairal (2016). We find that replacing the second layer kernel by a simple polynomial kernel of degree 3, $\kappa_2(u) = u^3$, leads to roughly the same test accuracy. By changing $\kappa_2$ to $\kappa_2(u) = u^2$, the test accuracy is only about 1% lower, while doing the same for the first layer decreases it by about 3%. The shallow kernel with a single non-linear convolutional layer (shown in the last line of Table 1) performs significantly worse. This suggests that the approximation properties described in Section 3.1 may not be sufficient for this task, while even a simple polynomial kernel of order 2 at the second layer may substantially improve things by capturing interactions, in a way that we describe below.

**Two-layers with a quadratic kernel.** Motivated by the above experiments, we now study the RKHS of a two-layer kernel $K_2(x, x') = \langle \Psi(x), \Psi(x') \rangle_{L^2(\Omega_2, \mathcal{H}_2)}$ with $\Psi$ as in (1) with $L = 2$, where the second-layer uses a quadratic kernel[2] on patches $k_2(z, z') = (\langle z, z' \rangle)^2$. An explicit feature map for $k_2$ is given by $\varphi_2(z) = z \otimes z$. Denoting by $\mathcal{H}$ the RKHS of $k_1$, the patches $z$ lie in $\mathcal{H}^{|S_2|}$, thus we may view $\varphi_2$ as a feature map into a Hilbert space $\mathcal{H}_2 = (\mathcal{H} \otimes \mathcal{H})^{|S_2| \times |S_2|}$ (by isomorphism to $\mathcal{H}^{|S_2|} \otimes \mathcal{H}^{|S_2|}$). The following result characterizes the RKHS of such a 2-layer convolutional kernel, showing that it consists of additive models of interaction functions in $\mathcal{H} \otimes \mathcal{H}$ on pairs of patches, with different spatial regularities on the interaction terms induced by the two pooling operations. (**Notation**: we use the notations $\text{diag}(M)[u] = M[u, u]$ for $M \in L^2(\Omega^2)$, $\text{diag}(x)[u, v] = \mathbb{1}\{u = v\}x[u]$ for $x \in L^2(\Omega)$, and $L_c$ is the translation operator $L_c x[u] = x[u - c]$.)

**Proposition 2** (RKHS of 2-layer CKN with quadratic $k_2$). *Let* $\Phi(x) = (\varphi_1(x_u) \otimes \varphi_1(x_v))_{u,v \in \Omega} \in L^2(\Omega^2, \mathcal{H} \otimes \mathcal{H})$. *The RKHS of* $K_2$ *when* $k_2(z, z') = (\langle z, z' \rangle)^2$ *consists of functions of the form*

$$f(x) = \sum_{p,q \in S_2} \langle G_{pq}, \Phi(x) \rangle = \sum_{p,q \in S_2} \sum_{u,v \in \Omega} G_{pq}[u, v](x_u, x_v),$$

*where* $G_{pq} \in L^2(\Omega^2, \mathcal{H} \otimes \mathcal{H})$ *obeys the constraints* $G_{pq} \in \text{Range}(E_{pq})$ *and* $\text{diag}((L_p A_1 \otimes L_q A_1)^{\dagger *} G_{pq}) \in \text{Range}(A_2^*)$. *Here,* $E_{pq} : L^2(\Omega_1) \to L^2(\Omega^2)$ *is a linear operator given by*

$$E_{pq}x = (L_p A_1 \otimes L_q A_1)^* \text{diag}(x). \tag{6}$$

*The squared RKHS norm* $\|f\|^2_{\mathcal{H}_{K_2}}$ *is then equal to the minimum over such decompositions of the quantity*

$$\sum_{p,q \in S_2} \|A_2^{\dagger *} \text{diag}((L_p A_1 \otimes L_q A_1)^{\dagger *} G_{pq})\|^2_{L^2(\Omega_2, \mathcal{H} \otimes \mathcal{H})}. \tag{7}$$

As discussed in the one-layer case, the inverses should be replaced by pseudo-inverses if needed, *e.g.*, when using downsampling. In particular, if $A_2^*$ is singular, the second constraint plays a similar role to the one-layer case. In order to understand the first constraint, we show in Figure 2 the outputs

---

[2]For simplicity we study the quadratic kernel instead of the homogeneous version used in the experiments, noting that it still performs well (78.0% instead of 79.4% on 10k examples).

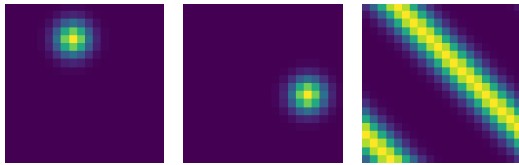

Figure 2: Display of the 2D response $E_{pq}x \in L^2(\Omega^2)$ of the operator in (6) for various 1D inputs $x \in L^2(\Omega)$. (left/center) Dirac inputs $x = \delta_u$ centered at two different locations $u$; (right) Constant input $x = \mathbf{1}$. The responses are localized on the $p - q$ diagonal, corresponding to interactions between two patches at distance around $p - q$. Here, we took $(p, q) = (4, 0)$, with a signal size $|\Omega| = 20$.

of $E_{pq}x$ for Dirac delta signals $x[v] = \delta_u[v]$. We can see that if the pooling filter $h_1$ has a small support of size $m$, then $G_{pq}[u - p, v - q]$ must be zero when $|u - v| > m$, which highlights that the functions in $G_{pq}$ may only capture interactions between pairs of patches where the (signed) distance between the first and the second is close to $p - q$.

The penalty then involves operators $L_p A_1 \otimes L_q A_1$, which may be seen as separable 2D convolutions on the "images" $G_{pq}[u, v]$. Then, if $z, z' \in \mathbb{R}^{p|S_1|}$ are two fixed patches, defining $g_{z,z'}[u, v] = G_{pq}[u, v](z, z')$, we have, assuming $A_1$ is invertible and symmetric,

$$(L_p A_1 \otimes L_q A_1)^{\dagger *} G_{pq}[u, v](z, z') = (A_1 \otimes A_1)^{-1} g_{z,z'}[u - p, v - q]$$
$$= \mathcal{F}_2^{-1} \operatorname{diag}(\mathcal{F}_2(h_1 \otimes h_1))^{-1} \mathcal{F}_2 g_{z,z'}[u - p, v - q],$$

where $\mathcal{F}_2 = \mathcal{F} \otimes \mathcal{F}$ is the 2D discrete Fourier transform. Thus, this penalizes the variations of $g_{z,z'}$ in both dimensions, encouraging the interaction functions to not rely too strongly on the specific positions of the two patches. This regularization is stronger when the spatial bandwidth of $h_1$ is large, since this leads to a more localized filter in the frequency domain, with stronger penalties on high frequencies. In addition to this 2D smoothness, the penalty in Proposition 2 also encourages smoothness along the $p - q$ diagonal of this resulting 2D image using the pooling operator $A_2$. This has a similar behavior to the one-layer case, where the penalty prevents the functions from relying too much on the absolute position of the patches. Since $A_2$ typically has a larger bandwidth than $A_1$, interaction functions $G_{pq}[u, u + r]$ are allowed to vary with $r$ more rapidly than with $u$. The regularity of the resulting "smoothed" interaction terms as a function of the input patches is controlled by the RKHS norm of the tensor product kernel $k_1 \otimes k_1$ as described in Appendix A.2.

**Extensions.** When using a polynomial kernel $k_2(z, z') = (\langle z, z' \rangle)^\alpha$ with $\alpha > 2$, we obtain a similar picture as above, with higher-order interaction terms. For example, if $\alpha = 3$, the RKHS contains functions with interaction terms of the form $G_{pqr}[u, v, w](x_u, x_v, x_w)$, with a penalty

$$\sum_{p,q,r \in S_2} \|A_2^{\dagger *} \operatorname{diag}((A_{1p} \otimes A_{1q} \otimes A_{1r})^{\dagger *} G_{pqr})\|_{L^2(\Omega_2, \mathcal{H}^{\otimes 3})}^2,$$

where $A_{1c} = L_c A_1$. Similarly to the quadratic case, the first-layer pooling operator encourages smoothness with respect to relative positions between patches, while the second-layer pooling penalizes dependence on the global location. One may extend this further to higher orders to capture more complex interactions, and our experiments suggest that a two-layer kernel of this form with a degree-4 polynomial at the second layer may achieve state-of-the-art accuracy for kernel methods on Cifar10 (see Table 2). We note that such fixed-order choices for $\kappa_2$ lead to convolutional kernels that lower-bound richer kernels with, *e.g.*, an exponential kernel at the second layer, in the Loewner order on positive-definite kernels. This imples in particular that the RKHS of these "richer" kernels also contains the functions described above. For more than two layers with polynomial kernels, one similarly obtains higher-order interactions, but with different regularization properties (see Appendix D).

## 4 GENERALIZATION PROPERTIES

In this section, we study generalization properties of the convolutional kernels studied in Section 3, and show improved sample complexity guarantees for architectures with pooling and small patches when the problem exhibits certain invariance properties.

**Learning setting.** We consider a non-parametric regression setting with data distribution $\rho$ over $(x, y)$, where the goal is to minimize $R(f) = \mathbb{E}_{(x,y)\sim\rho}[(y - f(x))^2]$. We denote by $f^* = \mathbb{E}_\rho[y|x] = \arg\min_f R(f)$ the regression function, and assume $f^* \in \mathcal{H}$ for some RKHS $\mathcal{H}$ with kernel $K$. Without any further assumptions on the kernel, we have the following generalization bound on the excess risk for the kernel ridge regression (KRR) estimator, denoted $\hat{f}_n$ (see Proposition 7 in Appendix E):

$$\mathbb{E}[R(\hat{f}_n) - R(f^*)] \leq C\|f^*\|_{\mathcal{H}}\sqrt{\frac{\tau_\rho^2\,\mathbb{E}_{x\sim\rho_X}[K(x,x)]}{n}}, \tag{8}$$

where $\rho_X$ is the marginal distribution of $\rho$ on inputs $x$, $C$ is an absolute constant, and $\tau_\rho^2$ is an upper bound on the conditional noise variance $\mathrm{Var}[y|x]$. We note that this $1/\sqrt{n}$ rate is optimal if no further assumptions are made on the kernel (Caponnetto & De Vito, 2007). The quantity $\mathbb{E}_{x\sim\rho_X}[K(x,x)]$ corresponds to the trace of the covariance operator, and thus provides a global control of eigenvalues through their sum, which will already highlight the gains that pooling can achieve. Faster rates can be achieved, *e.g.*, when assuming certain eigenvalue decays on the covariance operator, or when further restricting $f^*$. We discuss in Appendix F.2 how similar gains to those described in this section can extend to fast rate settings under specific scenarios.

**One-layer CKN with invariance.** As discussed in Section 3, the RKHS of 1-layer CKNs consists of sums of functions that are localized on patches, each belonging to the RKHS $\mathcal{H}$ of the patch kernel $k_1$. The next result illustrates the benefits of pooling when $f^*$ is translation invariant.

**Proposition 3** (Generalization for 1-layer CKN.). *Assume $f^*(x) = \sum_{u\in\Omega} g(x_u)$ with $g \in \mathcal{H}$ of minimal norm, and assume $\mathbb{E}_{x\sim\rho_X}[k_1(x_u, x_v)] \leq \sigma_{u-v}^2$ for some $(\sigma_r^2)_{r\in\Omega}$. For a 1-layer CKN $K_1$ with any pooling filter $h \geq 0$ with $\|h\|_1 = 1$, we have $\|f^*\|_{\mathcal{H}_{K_1}} = \sqrt{|\Omega|}\|g\|_{\mathcal{H}}$, and KRR satisfies*

$$\mathbb{E}\,R(\hat{f}_n) - R(f_*) \lesssim \frac{|\Omega|\|g\|_{\mathcal{H}}}{\sqrt{n}}\sqrt{\tau_\rho^2\sum_{r\in\Omega}\langle h, L_r h\rangle\sigma_r^2}. \tag{9}$$

The quantities $\sigma_r^2$ can be interpreted as auto-correlations between patches at distance $r$ from each other. Note that if $h$ is a Dirac filter, then $\langle h, L_r h\rangle = \mathbb{1}\{r = 0\}$ (recall $L_r h[u] = h[u - r]$), thus only $\sigma_0^2$ plays a role in the bound, while if $h$ is an average pooling filter, we have $\langle h, L_r h\rangle = 1/|\Omega|$, so that $\sigma_0^2$ is replaced by the average $\bar{\sigma}^2 := \sum_r \sigma_r^2/|\Omega|$. Natural signals commonly display a decay in their auto-correlation functions, suggesting that a similar decay may be present in $\sigma_r^2$ as a function of $r$. In this case, $\bar{\sigma}^2$ may be much smaller than $\sigma_0^2$, which in turn yields an *improved sample complexity guarantee* for learning such an $f^*$ with global pooling, by a factor up to $|\Omega|$ in the extreme case where $\sigma_r^2$ vanishes for $r \geq 1$ (since $\bar{\sigma}^2 = \sigma_0^2/|\Omega|$ in this case). In Appendix F.1, we provide simple models where this can be quantified. For more general filters, such as local averaging or Gaussian filters, and assuming $\sigma_r^2 \approx 0$ for $r \neq 0$, the bound interpolates between no pooling and global pooling through the quantity $\|h\|_2^2$. While this yields a worse bound than global pooling on invariant functions, such filters enable learning functions that are not fully invariant, but exhibit some smoothness along the translation group, more efficiently than with no pooling. It should also be noted that the requirement that $g$ belongs to an RKHS $\mathcal{H}$ is much weaker when the patches are small, as this typically implies that $g$ admits more than $p|S|/2$ derivatives, a condition which becomes much stronger as the patch size grows. In the fast rate setting that we study in Appendix F.2, this also leads to better rates that only depend on the dimension of the patch instead of the full dimension (see Theorem 8 in Appendix F.2).

**Two layers.** When using two layers with polynomial kernels at the second layer, we saw in Section 3 that the RKHS of CKNs consists of additive models of interaction terms of the order of the polynomial kernel used. The next proposition illustrates how pooling filters and patch sizes at the second layer may affect generalization on a simple target function consisting of order-2 interactions.

**Proposition 4** (Generalization for 2-layer CKN.). *Consider a 2-layer CKN $K_2$ with quadratic $k_2$, as in Proposition 2, and pooling filters $h_1, h_2$ with $\|h_1\|_1 = \|h_2\|_1 = 1$. Assume that $\rho_X$ satisfies $\mathbb{E}_{x\sim\rho_X}[k_1(x_u, x_{u'})k_1(x_v, x_{v'})] \leq \epsilon$ if $u \neq u'$ or $v \neq v'$, and $\leq 1$ otherwise. We have*

$$\mathbb{E}_{x\sim\rho_X}[K_2(x,x)] \leq |S_2|^2|\Omega|\left(\sum_v \langle h_2, L_v h_2\rangle\langle h_1, L_v h_1\rangle^2 + \epsilon\right). \tag{10}$$

As an example, consider $f^*(x) = \sum_{u,v} g(x_u, x_v)$ for $g \in \mathcal{H} \otimes \mathcal{H}$ of minimal norm. The following table illustrates the obtained generalization bounds $R(\hat{f}_n) - R(f^*)$ for KRR with various two-layer architectures ($\delta$: Dirac filter; $\mathbf{1}$: global average pooling):

| $h_1$ | $h_2$ | $|S_2|$ | $\|f^*\|_{K_2}$ | $\mathbb{E}_{x \sim \rho_X}[K_2(x,x)]$ | Bound ($\epsilon = 0, \tau_\rho^2 = 1$) |
|---|---|---|---|---|---|
| $\delta$ | $\delta$ | $|\Omega|$ | $|\Omega|\|g\|$ | $|\Omega|^3 + \epsilon|\Omega|^3$ | $\|g\||\Omega|^{2.5}/\sqrt{n}$ |
| $\delta$ | $\mathbf{1}$ | $|\Omega|$ | $|\Omega|\|g\|$ | $|\Omega|^2 + \epsilon|\Omega|^3$ | $\|g\||\Omega|^2/\sqrt{n}$ |
| $\mathbf{1}$ | $\mathbf{1}$ | $|\Omega|$ | $\sqrt{|\Omega|}\|g\|$ | $|\Omega| + \epsilon|\Omega|^3$ | $\|g\||\Omega|/\sqrt{n}$ |
| $\mathbf{1}$ | $\delta$ or $\mathbf{1}$ | $1$ | $\sqrt{|\Omega|}\|g\|$ | $|\Omega|^{-1} + \epsilon|\Omega|$ | $\|g\|/\sqrt{n}$ |

The above result shows that the two-layer model allows for a much wider range of behaviors than the one-layer case, between approximation (through the norm $\|f^*\|_{K_2}$) and estimation (through $\mathbb{E}_{x \sim \rho_X}[K_2(x,x)]$), depending on the choice of architecture. Choosing the right architecture may lead to large improvements in sample complexity when the target functions has a specific structure, for instance here by a factor up to $|\Omega|^{2.5}$. In Appendix F.1, we discuss simple possible models where we may have a small $\epsilon$. Note that choosing filters that are less localized than Dirac impulses, but more than global average pooling, will again lead to different "variance" terms (10), while providing more flexibility in terms of approximation compared to global pooling. This result may be easily extended to higher-order polynomials at the second layer, by increasing the exponents on $|S_2|$ and $\langle h_1, L_v h_1 \rangle$ to the degree of the polynomial. Other than the gains in sample complexity due to pooling, the bound also presents large gains compared to a "fully-connected" architecture, as in the one-layer case, since it only grows with the norm of a local interaction function in $\mathcal{H} \otimes \mathcal{H}$ that depends on two patches, which may then be small even when this function has low smoothness.

## 5 Numerical Experiments

In this section, we provide additional experiments illustrating numerical properties of the convolutional kernels considered in this paper. We focus here on the Cifar10 dataset, and on CKN architectures based on the exponential kernel. Additional results are given in Appendix B.

**Experimental setup on Cifar10.** We consider classification on Cifar10 dataset, which consists of 50k training images and 10k test images with 10 different output categories. We pre-process the images using a whitening/ZCA step at the patch level, which is commonly used for such kernels on images (Mairal, 2016; Shankar et al., 2020; Thiry et al., 2021). This may help reduce the effective dimensionality of patches, and better align the dominant eigen directions to the target function, a property which may help kernel methods (Ghorbani et al., 2020). Our convolutional kernel evaluation code is written in C++ and leverages the Eigen library for hardware-accelerated numerical computations. The computation of kernel matrices is distributed on up to 1000 cores on a cluster consisting of Intel Xeon processors. Computing the full Cifar10 kernel matrix typically takes around 10 hours when running on all 1000 cores. Our results use kernel ridge regression in a one-versus-all approach, where each class uses labels $0.9$ for the correct label and $-0.1$ for the other labels. We report the test accuracy for a fixed regularization parameter $\lambda = 10^{-8}$ (we note that the performance typically remains the same for smaller values of $\lambda$). The exponential kernel always refers to $\kappa(u) = e^{\frac{1}{\sigma^2}(u-1)}$ with $\sigma = 0.6$. Code is available at `https://github.com/albietz/ckn_kernel`.

**Varying the kernel architecture.** Table 2 shows test accuracies for different architectures compared to Table 1, including 3-layer models and 2-layer models with larger patches. In both cases, the full models with exponential kernels outperform the 2-layer architecture of Table 1, and provide comparable accuracy to the Myrtle10 kernel of Shankar et al. (2020), with an arguably simpler architecture. We also see that using degree-3 or 4 polynomial kernels at the second second layer of the two-layer model essentially provides the same performance to the exponential kernel, and that degree-2 at the second and third layer of the 3-layer model only results in a 0.3% accuracy drop. The two-layer model with degree-2 at the second layer loses about 1% accuracy, suggesting that certain Cifar10 images may require capturing interactions between at least 3 different patches in the image for good classification, though even with only second-order interactions, these models significantly outperform single-layer models. While these results are encouraging, computing such kernels is prohibitively costly, and we found that applying the Nyström approach of Mairal (2016) to these

Table 2: Cifar10 test accuracy for two-layer architectures with larger second-layer patches, or three layer architectures. $\kappa$ denote the patch kernels used at each layer, 'conv' the patch sizes, and 'pool' the downsampling factors for Gaussian pooling filters. We include the Myrtle10 convolutional kernel Shankar et al. (2020), which consists of 10 layers including Exp kernels on 3x3 patches and 2x2 average pooling.

| $\kappa$ | conv | pool | Test acc. (10k) | Test acc. (50k) |
|---|---|---|---|---|
| (Exp,Exp) | (3,5) | (2,5) | 81.1% | 88.3% |
| (Exp,Poly4) | (3,5) | (2,5) | 81.3% | 88.3% |
| (Exp,Poly3) | (3,5) | (2,5) | 81.1% | 88.2% |
| (Exp,Poly2) | (3,5) | (2,5) | 80.1% | 87.4% |
| (Exp,Exp,Exp) | (3,3,3) | (2,2,2) | 80.7% | 88.2% |
| (Exp,Poly2,Poly2) | (3,3,3) | (2,2,2) | 80.5% | 87.9% |
| Myrtle10 Shankar et al. (2020) | - | - | - | 88.2% |

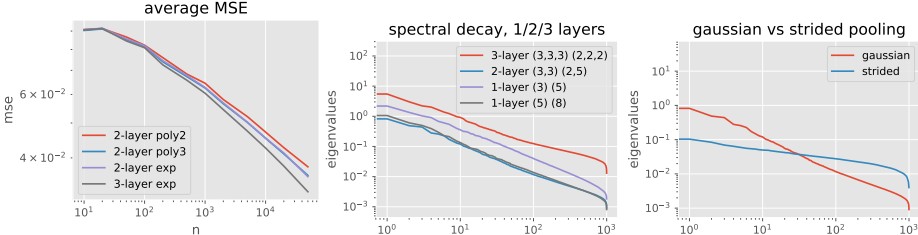

Figure 3: (left) Mean squared error of kernel ridge regression on Cifar10 for different kernels with 3x3 patches as a function of sample size (averaged over the 10 classes). (center) Eigenvalue decays of kernel matrices (with Exp kernels) on 1000 Cifar images, for different depths, patch sizes (3 or 5) and pooling sizes. (right) decays for 2-layer architecture, Gaussian pooling filter vs strided convolutions (*i.e.*, no pooling). The plots illustrate that pooling is essential for reducing effective dimensionality.

kernels with more layers or larger patches requires larger models than for the architecture of Table 1 for a similar accuracy. Figure 3(left) shows learning curves for different architectures, with slightly better convergence rates for more expressive models involving higher-order kernels or more layers; this suggests that their approximation properties may be better suited for these datasets.

**Role of pooling.** Figure 3 shows the spectral decays of the empirical kernel matrix on 1000 Cifar images, which may help assess the "effective dimensionality" of the data, and are related to generalization properties (Caponnetto & De Vito, 2007). While multi-layer architectures with pooling seem to provide comparable decays for various depths, removing pooling leads to significantly slower decays, and hence much larger RKHSs. In particular, the "strided pooling" architecture (*i.e.*, with Dirac pooling filters and downsampling) shown in Figure 3(right), which resembles the kernel considered in (Scetbon & Harchaoui, 2020), obtains less than 40% accuracy on 10k examples. This suggests that the regularization properties induced by pooling, studied in Section 3, are crucial for efficient learning on these problems, as shown in Section 4. Appendix B provides more empirics on different pooling configurations.

## 6 DISCUSSION AND CONCLUDING REMARKS

In this paper, we studied approximation and generalization properties of convolutional kernels, showing how multi-layer models with convolutional architectures may effectively break the curse of dimensionality on problems where the input consists of high-dimensional natural signals, by modeling localized functions on patches and interactions thereof. We also show how pooling induces additional smoothness constraints on how interaction terms may or may not vary with global and relative spatial locations. An important question for future work is how optimization of deep convolutional networks may further improve approximation properties compared to what is captured by the kernel regime presented here, for instance by selecting well-chosen convolution filters at the first layer, or interaction patterns in subsequent layers, perhaps in a hierarchical manner.

ACKNOWLEDGMENTS

The author would like to thank Francis Bach, Alessandro Rudi, Joan Bruna, and Julien Mairal for helpful discussions.

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

## A    FURTHER BACKGROUND

This section provides further background on the problem of approximation of functions defined on signals, as well as on the kernels considered in the paper. We begin by introducing and motivating the problem of learning functions defined on signals such as images, which captures tasks such as image classification where deep convolutional networks are predominant. We then recall properties of dot-product kernels and kernel tensor products, which are key to our study of approximation.

### A.1    NATURAL SIGNALS AND CURSE OF DIMENSIONALITY

We consider learning problems consisting of labeled examples $(x, y) \sim \rho$ from a data distribution $\rho$, where $x$ is a discrete signal $x[u]$ with $u \in \Omega$ denoting the position (*e.g.*, pixel location in an image) in a domain $\Omega$, $x[u] \in \mathbb{R}^p$ (*e.g.*, $p = 3$ for RGB pixels), and $y \in \mathbb{R}$ is a target label. In a non-parametric setup, statistical learning may be framed as trying to approximate the regression function

$$f^*(x) = \mathbb{E}_\rho[y|x]$$

using samples from the data distribution $\rho$. If $f^*$ is only assumed to be Lipschitz, learning requires a number of samples that scales exponentially in the dimension (see, *e.g.*, von Luxburg & Bousquet (2004); Wainwright (2019)), a phenomenon known as the *curse of dimensionality*. In the case of natural signals, the dimension $d = p|\Omega|$ scales with the size of the domain $|\Omega|$ (*e.g.*, the number of pixels), which is typically very large and thus makes this intractable. One common way to alleviate this is to assume that $f^*$ is smooth, however the order of smoothness typically needs to be of the order of the dimension in order for the problem to become tractable, which is a very strong assumption here when $d$ is very large. This highlights the need for more structured assumptions on $f^*$ which may help overcome the curse of dimensionality.

**Insufficiency of invariance and stability.** Two geometric properties that have been successful for studying the benefits of convolutional architectures are (near-)translation invariance and stability to deformations. Various works have shown that certain convolutional models $f$ yield good invariance and stability Mallat (2012); Bruna & Mallat (2013); Bietti & Mairal (2019a), in the sense that when $\tilde{x}$ is a translation or a small deformation of $x$, then $|f(\tilde{x}) - f(x)|$ is small. Nevertheless, one can show that for band-limited signals (such as discrete signals), $\|\tilde{x} - x\|_2$ can be controlled in a similar way (though with worse constants, see (Wiatowski & Bölcskei, 2018, Proposition 5)), so that Lipschitz functions on such signals obey such stability properties. Thus, deformation stability is not a much stronger assumption than Lipschitzness, and is insufficient by itself to escape the curse of dimensionality.

**Spatial localization.** One successful strategy for learning image recognition models which predates deep learning is to rely on simple aggregations of local features. These may be extracted using hand-crafted procedures (Lowe, 1999; Sánchez et al., 2013; Jégou et al., 2011), or using learned feature extractors, either through learned filters in the early layers of a CNN (Zeiler & Fergus, 2014), or other procedures (*e.g.*, Thiry et al. (2021)). One simplified example that encodes such a prior is if the target function $f^*$ only depends on the input image through a localized part of the input such as a patch $x_u = (x[u + v])_{v \in S} \in \mathbb{R}^{p|S|}$, where $S$ is a small box centered around 0, that is, $f^*(x) = g^*(x_u)$. Then, if $g^*$ is assumed to be Lipschitz, we would like a sample complexity that only scales exponentially in the dimension of a patch $p|S|$, which is much smaller than dimension of the entire image $p|\Omega|$. This is indeed the case if we use a kernel defined on such patches, such as

$$K(x, x') = \sum_u k(x_u, x'_u),$$

where $k$ is a "simple" kernel such as a dot-product kernel, as discussed in Appendix C. In contrast, if $K$ is a dot-product kernel on the entire image, corresponding to an infinite-width limit of a fully-connected network, then approximation is more difficult and is generally cursed by the full dimension (see Appendix C). While some models of wide fully-connected networks provide some adaptivity to low-dimensional structures such as the variables in a patch (Bach, 2017a), no tractable algorithms are currently known to achieve such behavior provably, and it is reasonable to instead encode such prior information in a convolutional architecture.

**Modeling interactions.** Modeling interactions between elements of a system at different scales, possibly hierarchically, is important in physics and complex systems, in order to efficiently handle systems with large numbers of variables (Beylkin & Mohlenkamp, 2002; Hackbusch & Kühn, 2009). As an example, one may consider target functions $f^*(x)$ that consist of interaction functions of the form $g(x_p, x_q)$, where $p, q$ denote locations of the corresponding patches, and higher-order interactions may also be considered. In the context of image recognition, while functions of a single patch may capture local texture information such as edges or color, such an interaction function may also respond to specific spatial configurations of relevant patches, which could perhaps help identify properties related to the "shape" of an object, for instance. If such functions $g$ are too general, then the curse of dimensionality may kick in again when one considers more than a handful of patches. Certain idealized models of approximation may model such interactions more efficiently through hierarchical compositions (*e.g.*, Poggio et al. (2017)) or tensor decompositions (Cohen & Shashua, 2016; 2017), though no tractable algorithms are known to find such models. In this work, we tackle this in a tractable way using multi-layer convolutional kernels. We show that they can model interactions through kernel tensor products, which define functional spaces that are typically much smaller and more structured than for a generic kernel on the full vector $(x_p, x_q)$.

A.2 Dot-Product Kernels and their Tensor Products

In this section, we review some properties of dot-product kernels, their induced RKHS and regularization properties. We then recall the notion of tensor product of kernels, which allows us to describe the RKHS of products of kernels in terms of that of individual kernels.

**Dot-product kernels.** The rotation-invariance of dot-product kernels provides a natural description of their RKHS in terms of harmonic decompositions of functions on the sphere using spherical harmonics (Smola et al., 2001; Bach, 2017a). This leads to natural connections with regularity

properties of functions defined on the sphere. For instance, if the kernel integral operator on $L^2(\mathbb{S}^{d-1})$ has a polynomially decaying spectral decay, as is the case for kernels arising from the ReLU activation (Bach, 2017a; Bietti & Mairal, 2019b), then the RKHS contains functions $g \in L^2(\mathbb{S}^{d-1})$ with an RKHS norm equivalent to

$$\|\Delta_{\mathbb{S}^{d-1}}^{\beta/2} g\|_{L^2(\mathbb{S}^{d-1})}, \tag{11}$$

for some $\beta$ that depends on the decay exponent and must be larger than $(d-1)/2$, with $\Delta_{\mathbb{S}^{d-1}}$ the Laplace-Beltrami operator on the sphere. This resembles a Sobolev norm of order $\beta$, and the RKHS contains functions with bounded derivatives up to order $\beta$. When $d$ is small (*e.g.*, at the first layer with small images patches), the space contains functions that need not be too regular, and may thus be quite discriminative, while for large $d$ (*e.g.*, for a fully-connected network), the functions must be highly smooth in order to be in the RKHS, and large norms are necessary to approach non-smooth functions. For kernels with decays faster than polynomial, such as the Gaussian kernel, the RKHS contains smooth functions, but may still provide good approximation to non-smooth functions, particularly with small $d$ and when using small bandwidth parameters. The homogeneous case (2) leads to functions $f(x) = \|x\|g(\frac{x}{\|x\|})$ with $g$ defined on the sphere, with a norm given by the same penalty (11) on the function $g$ (Bietti & Mairal, 2019b).

**Kernel tensor products.**   For more than one layer, the convolutional kernels we study in Section 3 can be expressed in terms of products of kernels on patches, of the form

$$K((x_1, \ldots, x_m), (x'_1, \ldots, x'_m)) = \prod_{j=1}^{m} k(x_j, x'_j), \tag{12}$$

where $x_1, \ldots, x_m, x'_1, \ldots, x'_m \in \mathbb{R}^d$ are patches which may come from different signal locations. If $\varphi : \mathbb{R}^d \to \mathcal{H}$ is the feature map into the RKHS $\mathcal{H}$ of $k$, then

$$\psi(x_1, \ldots, x_m) = \varphi(x_1) \otimes \cdots \otimes \varphi(x_m)$$

is a feature map for $K$, and the corresponding RKHS, denoted $\mathcal{H}^{\otimes m} = \mathcal{H} \otimes \cdots \otimes \mathcal{H}$, contains all functions

$$f(x_1, \ldots, x_m) = \sum_{i=1}^{n} g_{i,1}(x_1) \ldots g_{i,m}(x_m),$$

for some $n$, with $g_{i,m} \in \mathcal{H}$ for $i \in [n]$ and $j \in [m]$ (see,*e.g.*, (Wainwright, 2019, Section 12.4.2) for a precise construction). The resulting RKHS is often much smaller than for a more generic kernel on $\mathbb{R}^{d \times m}$; for instance, if $\mathcal{H}$ is a Sobolev space of order $\beta$ in dimension $d$, then $\mathcal{H}^{\otimes m}$ is much smaller than the Sobolev space of order $\beta$ in $d \times m$ dimensions, and corresponds to stronger, mixed regularity conditions (see, *e.g.*, Bach, 2017b; Sickel & Ullrich, 2009, for the $d = 1$ case). This can yield improved generalization properties if the target function has such a structure (Lin, 2000). Kernels of the form (12) and sums of such kernels have been useful tools for avoiding the curse of dimensionality by encoding interactions between variables that are relevant to the problem at hand (Wahba, 1990, Chapter 10). In what follows, we show how patch extraction and pooling operations shape the properties of such interactions between patches in convolutional kernels through additional spatial regularities.

# B  ADDITIONAL EXPERIMENTS

In this section, we provide additional experiments to those presented in Section 5, using different patch kernels, patch sizes, pooling filters, preprocessings, and datasets.

**Three-layer architectures with different patch kernels.**   Table 3 provides more results on 3-layer architectures compared to Table 2, including different changes in the degrees of polynomial kernels at the second and third layer. In particular we see that the architecture with degree-2 kernels at both layers, which captures interactions of order 4, also outperforms the simpler ones using degree-4 kernels at either layer, suggesting that a deeper architecture may better model relevant interactions terms on this problem.

Table 3: Cifar10 test accuracy with 3-layer convolutional kernels with 3x3 patches and pooling/downsampling sizes [2,2,2], with different choices of patch kernels $\kappa_1$, $\kappa_2$ and $\kappa_3$. The last model is similar to a 1-layer convolutional kernel. Due to high computational cost, we use 10k training images instead of the full training set (50k images) in most cases.

| $\kappa_1$ | $\kappa_2$ | $\kappa_3$ | Test ac. (10k) | Test ac. (50k) |
|---|---|---|---|---|
| Exp | Exp | Exp | 80.7% | **88.2%** |
| Exp | Poly2 | Poly2 | 80.5% | 87.9% |
| Exp | Poly4 | Lin | 80.2% | - |
| Exp | Lin | Poly4 | 79.2% | - |
| Exp | Lin | Lin | 74.1% | - |

Table 4: Cifar10 test accuracy on 10k examples with 2-layer convolutional kernels with 3x3 patches at the first layer, pooling/downsampling sizes [2,5] and patch kernels [Exp,Poly2], with different patch sizes at the second layer.

| $|S_2|$ | 1x1 | 3x3 | 5x5 | 7x7 | 9x9 | 11x11 |
|---|---|---|---|---|---|---|
| Test acc. (10k) | 76.3% | 79.4% | 80.1% | 80.1% | 80.1% | 79.9% |

**Varying the second layer patch size.** Table 4 shows the variations in test performance when changing the size of the second patches at the second layer. We see that intermediate sizes between 3x3 and 9x9 work best, but that performance degrades when using patches that are too large or too small. For very large patches, this may be due to the large variance in (10), or perhaps instability (Bietti & Mairal, 2019a). For $|S_2| = $1x1, note that while pooling after the first layer allows even 1x1 patches to capture interactions across different input image patches, these may be limited to short range interactions when the pooling filter is localized (see Proposition 2), which may limit the expressivity of the model.

Table 5: Cifar10 test accuracy with patch kernels that are either arc-cosine kernels (denoted ReLU) or polynomial kernels. The 2-layer architectures use 3x3 patches and [2,5] downsampling/pooling as in Table 1. "ReLU-NTK" indicates that we consider the neural tangent kernel for a ReLU network with similar architecture, instead of the conjugate kernel.

| $\kappa_1$ | $\kappa_2$ | Test ac. (10k) | Test ac. (50k) |
|---|---|---|---|
| ReLU | ReLU | 78.5% | 86.6% |
| ReLU-NTK | ReLU-NTK | 79.2% | 87.2% |
| ReLU | Poly2 | 77.2% | - |
| ReLU | Lin | 71.5% | - |

**Arc-cosine kernel.** In Table 5, we consider 2-layer convolutional kernels with a similar architecture to those considered in Table 1, but where we use arc-cosine kernels arising from ReLU activations instead of the exponential kernel used in Section 5, given by

$$\kappa(u) = \frac{1}{\pi}(u \cdot (\pi - \arccos(u)) + \sqrt{1 - u^2}).$$

The obtained convolutional kernel then corresponds to the conjugate kernel or NNGP kernel arising from an infinite-width convolutional network with the ReLU activation (Daniely et al., 2016; Garriga-Alonso et al., 2019; Novak et al., 2019). We may also consider the neural tangent kernel (NTK) for the same architecture, which additionally involves arc-cosine kernels of degree 0, which correspond to random feature kernels for step activations $u \mapsto \mathbb{1}\{u \geq 0\}$. We find that the NTK performs slightly better than the conjugate kernel, but both kernels achieve lower accuracy compared to the Exponential kernel shown in Table 1. Nevertheless, we observe a similar pattern regarding the use of polynomial kernels at the second layer, namely, the drop in accuracy is much smaller when using a quadratic kernel compared to a linear kernel, suggesting that non-linear kernels on top of the first layer, and the interactions they may capture, are crucial on this dataset for good accuracy.

Table 6: Cifar10 test accuracy for one-layer architectures with larger patches of size 6x6, exponential kernels, and different downsampling/pooling sizes (using Gaussian pooling filters with bandwidth and size of filters proportional to the downsampling factor). The results are for 10k training samples.

| Pooling | 2 | 4 | 6 | 8 | 10 |
|---|---|---|---|---|---|
| Test acc. (10k) | 67.6% | 73.3% | 75.5% | 75.8% | 75.5% |

**One-layer architectures and larger initial patches.** Table 6 shows the accuracy for one-layer convolutional kernels with 6x6 patches[3] and various pooling sizes, with a highest accuracy of 75.8% for a pooling size of 8. While this improves on the accuracy obtained with 3x3 patches (slightly above 74% for the architectures in Tables 1 and 3 with a single non-linear kernel at the first layer), these accuracies remain much lower than those achieved by two-layer architectures with even quadratic kernels at the second layer. While using larger patches may allow capturing patterns that are less localized compared to small 3x3 patches, the neighborhoods that they model need to remain small in order to avoid the curse of dimensionality when using dot-product kernels, as discussed in Section 2. Instead, the multi-layer architecture may model information at larger scales with a much milder dependence on the size of the neighborhood, thanks to the structure imposed by tensor product kernels (see Section A.2) and the additional regularities induced by pooling.

We also found that larger patches at the first layer may hurt performance in multi-layer models: when considering the architecture of Table 1 with exponential kernels, using 5x5 patches instead of 3x3 at the first layer yields an accuracy of 79.6% instead of 80.5% on Cifar10 when training on the same 10k images. This again reflects the benefits of using small patches at the first layer for allowing better approximation on small neighborhoods, while modeling larger scales using interaction models according to the structure of the architecture. We note nevertheless that for standard deep networks, larger patches are often used at the first layer (*e.g.*, He et al., 2016), as the feature selection capabilities of SGD may alleviate the dependence on dimension, *e.g.*, by finding Gabor-like filters.

**Gaussian vs average pooling.** Table 7 shows the differences in performance between two or three layer architectures considered in Table 2, when Gaussian pooling filters are replaced by average pooling filters. For both architectures considered, average pooling leads to a significant performance drop. This suggests that one may need deeper architectures in order for such average pooling filters to work well, as in (Shankar et al., 2020), either with multiple 3x3 convolutional layers before applying pooling, or by applying multiple average pooling layers in a row as in certain Myrtle kernels. Note that iterating multiple average pooling layers in a row is equivalent to using a larger and more smooth pooling filter (with one more order of smoothness at each layer), which may then be more comparable to our Gaussian pooling filters.

Table 7: Gaussian vs average pooling for two models from Table 2.

| Model | Gaussian | Average |
|---|---|---|
| $\kappa$: (Exp,Exp), conv: (3,5), pool: (2,5) | 88.3% | 75.9% |
| $\kappa$: (Exp,Exp,Exp), conv: (3,3,3), pool: (2,2,2) | 88.2% | 72.4% |

Table 8: SVHN test accuracy for a two-layer convolutional kernel network with Nyström approximation (Mairal, 2016) with patch size 3x3, pooling sizes [2,5], and filters [256, 4096].

| $\kappa_1$ | $\kappa_2$ | Test acc. (full with Nyström) |
|---|---|---|
| Exp | Exp | 89.5% |
| Exp | Poly3 | 89.3% |
| Exp | Poly2 | 88.6% |
| Poly2 | Exp | 87.1% |
| Poly2 | Poly2 | 86.6% |
| Exp | Lin | 78.5% |

---

[3]Note that in this case the ZCA/whitening step is applied on these larger 6x6 patches.

**SVHN dataset.** We now consider the SVHN dataset, which consists of 32x32 images of digits from Google Street View images, 73 257 for training and 26 032 for testing. Due to the larger dataset size, we only consider the kernel approximation approach of Mairal (2016) based on the Nyström method, which projects the patch kernel feature maps at each layer to finite-dimensional subspaces generated by a set of anchor points (playing the role of convolutional filters), themselves computed via a K-means clustering of patches.[4] We train one-versus-all classifiers on the resulting finite-dimensional representations using regularized ERM with the squared hinge loss, and simply report the best test accuracy over a logarithmic grid of choices for the regularization parameter, ignoring model selection issues in order to assess approximation properties. We use the same ZCA preprocessing as on Cifar10 and the same architecture as in Table 1, with a relatively small number of filters (256 at the first layer, 4096 at the second layer, leading to representations of dimension 65 536), noting that the accuracy can further improve when increasing this number. Our observations are similar to those for the Cifar10 dataset: using a degree-3 polynomial kernel at the second layer reaches very similar accuracy to the exponential kernel; using a degree-2 polynomial leads to a slight drop, but a smaller drop than when making this same change at the first layer; using a linear kernel at the second layer leads to a much larger drop. This again highlights the importance of using non-linear kernels on top of the first layer in order to capture interactions at larger scales than the scale of a single patch.

**Local versus global whitening.** Recall that our pre-processing is based on a patch-level whitening or ZCA on each image, following Mairal (2016). In practice, this is achieved by whitening extracted patches from each image, and reconstructing the image from whitened patches via averaging. In contrast, other approaches use global whitening of the entire image Lee et al. (2020); Shankar et al. (2020). For the 2-layer model shown in Table 2 with 5x5 patches at the second layer, we found global ZCA to provide significantly worse performance, with a drop from 88.3% to about 80%.

**Finite networks and comparison to Shankar et al. (2020).** The work Shankar et al. (2020) introduces Myrtle kernels but also consider similar architectures for usual CNNs with finite-width, trained with stochastic gradient descent. Obtaining competitive architectures for the finite-width case is not the goal of our work, which focuses on good architectures for the kernel setup, yet it remains interesting to consider this question. In the case of Shankar et al. (2020), training the finite-width networks yields better accuracy compared to their "infinite-width" kernel counterparts, a commonly observed phenomenon which may be due to better "adaptivity" of optimization algorithms compared to kernel methods, which have a fixed representation and thus may not learn representations adapted to the data (see, *e.g.*, Allen-Zhu & Li, 2020; Bach, 2017a; Chizat et al., 2019). Nevertheless, we found that for the two-layer architecture considered in Table 1, which has many fewer layers compared to the Myrtle architectures of Shankar et al. (2020), using a finite-width ReLU network yields poorer performance compared to the kernel (around 83% at best, compared to 87.9%). This may suggest that for convolutional networks, deeper networks may have additional advantages when using optimization algorithms, in terms of adapting to possibly relevant structure of the problem, such as hierarchical representations (see, *e.g.*, Allen-Zhu & Li (2020); Chen et al. (2020); Poggio et al. (2017) for theoretical justifications of the benefits of depth in non-kernel regimes).

## C    COMPLEXITY OF SPATIALLY LOCALIZED FUNCTIONS

In this section, we briefly elaborate on our discussion in Section A.1 on how simple convolutional structure may improve complexity when target functions are spatially localized. We assume $f^*(x) = g^*(x_u)$ with $x_u = (x[u+v])_{v \in S} \in \mathbb{R}^{p|S|}$ a patch of size $|S|$, where $g^*$ is a Lipschitz function.

If we define the kernel $K_u(x, x') = k(x_u, x'_u)$, where $k$ is a dot-product kernel arising from a one-hidden layer network with positively-homogeneous activation such as the ReLU, and further assume patches to be bounded and $g^*$ to be bounded, then the uniform approximation error bound of Bach (2017a, Proposition 6) together with a simple $O(1/\sqrt{n})$ Rademacher complexity bound on estimation error shows that we may achieve a generalization bound with a rate that only depends on the patch dimension $p|S|$ rather than $p|\Omega|$ in this setup (*i.e.*, a sample complexity that is exponential in $p|S|$, which is much smaller than $p|\Omega|$).

---

[4] We use the PyTorch implementation available at `https://github.com/claying/CKN-Pytorch-image`.

If we consider the kernel $K(x, x') = \sum_{u \in \Omega} k(x_u, x'_u)$, the RKHS contains all functions in the RKHS of $K_u$ for all $u \in \Omega$, with the same norm (this may be seen as an application of Theorem 6 with a feature map given by concatenating the kernel maps of each $K_u$), so that we may achieve the same approximation error as above, and thus a similar generalization bound that is not cursed by dimension. This kernel also allows us to obtain similar generalization guarantees when $f^*$ consists of linear combinations of such spatially localized functions on different patches within the image.

In contrast, when using a similar dot-product kernel on the full signal, corresponding to using a fully-connected network in a kernel regime, one may construct functions $f^*(x) = g^*(x_u)$ with $g^*$ Lipschitz where an RKHS norm that is exponentially large in the (full) dimension $p|\Omega|$ is needed for a small approximation error (see Bach, 2017a, Appendix D.5).

Related to this, Malach & Shalev-Shwartz (2021) show a separation in the different setting of learning certain parity functions on the hypercube using gradient methods; their upper bound for convolutional networks is based on a similar kernel regime as above. We note that kernels that exploit such a localized structure have also been considered in the context of structured prediction for improved statistical guarantees (Ciliberto et al., 2019).

## D    EXTENSIONS TO MORE LAYERS

In this section, we study the RKHS for convolutional kernels with more than 2 convolutional layers, by considering the simple example of a 3-layer convolutional kernel $K_3$ defined by the feature map

$$\Psi(x) = A_3 M_3 P_3 A_2 M_2 P_2 A_1 M_1 P_1 x,$$

with quadratic kernels at the second and third layer, *i.e.*, $k_2(z, z') = (\langle z, z' \rangle)^2$ and $k_3(z, z') = (\langle z, z' \rangle)^2$. By isomorphism, we may consider the sequence of Hilbert spaces $\mathcal{H}_\ell$ to be $\mathcal{H}_1 = \mathcal{H}$, $\mathcal{H}_2 = (\mathcal{H} \otimes \mathcal{H})^{|S_2| \times |S_2|}$, and $\mathcal{H}_3 = (\mathcal{H}^{\otimes 4})^{(|S_3| \times |S_2| \times |S_2|)^2}$. For some domain $\Omega$, we define the operators $\mathrm{diag}_2 : L^2(\Omega^4) \to L^2(\Omega^2)$ and its adjoint $\mathrm{diag}_2 : L^2(\Omega^2) \to L^2(\Omega^4)$ by

$$\mathrm{diag}_2(M)[u, v] = M[u, u, v, v] \quad \text{for } M \in L^2(\Omega^4)$$

$$\mathrm{diag}_2(M)[u_1, u_2, u_3, u_4] = \mathbb{1}\{u_1 = u_2\} \mathbb{1}\{u_3 = u_4\} M[u_1, u_3] \quad \text{for } M \in L^2(\Omega^2).$$

We may then describe the RKHS as follows.

**Proposition 5** (RKHS of 3-layer CKN with quadratic $k_{2/3}$). *The RKHS of $K_3$ when $k_2$ and $k_3$ are quadratic kernels $(\langle \cdot, \cdot \rangle)^2$ consists of functions of the form*

$$f(x) = \sum_{\alpha \in (S_3 \times S_2 \times S_2)^2} \sum_{u_1, u_2, u_3, u_v \in \Omega} G_\alpha[u_1, u_2, u_3, u_4](x_{u_1}, x_{u_2}, x_{u_3}, x_{u_4}), \qquad (13)$$

*where $G_\alpha \in L^2(\Omega^4, \mathcal{H}^{\otimes 4})$ obeys the constraint*

$$G_\alpha \in \mathrm{Range}(E_\alpha), \qquad (14)$$

*where the linear operator $E_\alpha : L^2(\Omega_3) \to L^2(\Omega^4)$ for $\alpha = (p, q, r, p', q', r')$ (with $p, p' \in S_3$ and $q, r, q', r' \in S_2$) is defined by*

$$E_\alpha x = A_{1,\alpha}^* \mathrm{diag}_2(A_{2,\alpha}^* \mathrm{diag}(A_{3,\alpha}^* x)).$$

*The operators $A_{1,\alpha}$ and $A_{2,\alpha}$ denote:*

$$A_{1,\alpha} = L_q A_1 \otimes L_r A_1 \otimes L_{q'} A_1 \otimes L_{r'} A_1$$
$$A_{2,\alpha} = L_p A_2 \otimes L_{p'} A_2.$$

*The squared RKHS norm $\|f\|_{\mathcal{H}_{K_3}}^2$ is then equal to the minimum over decompositions (13) of the quantity*

$$\sum_\alpha \|A_3^{\dagger*} \mathrm{diag}(A_{2,\alpha}^{\dagger*} \mathrm{diag}_2(A_{1,\alpha}^{\dagger*} G_\alpha))\|_{L^2(\Omega_3)}^2. \qquad (15)$$

The constraint (14) and penalty (15) resemble the corresponding constraint/penalty in the two-layer case for an order-4 polynomial kernel at the second layer, but provide more structure on the

interactions, using a multi-scale structure that may model interactions between certain pairs of patches $((x_{u_1}, x_{u_2})$ and $(x_{u_3}, x_{u_4})$ in (13)) more strongly than those between all four patches. In addition to localizing the interactions $G_\alpha$ around certain diagonals, the kernel also promotes spatial regularities: assuming that the spatial bandwidths of $A_\ell$ increase with $\ell$, the functions $(u, v, w_1, w_2) \mapsto G_\alpha[u, u + w_1, u + v, u + v + w_2]$ may vary quickly with $w_1$ or $w_2$ (distances between patches in each of the two pairs), but should vary more slowly with $v$ (distance between the two pairs) and even more slowly with $u$ (a global position).

## E  PROOFS

We recall the following result about reproducing kernel Hilbert spaces, which characterizes the RKHS of kernels defined by explicit Hilbert space features maps (see, *e.g.*, Saitoh, 1997, §2.1).

**Theorem 6** (RKHS from explicit feature map). *Let $H$ be some Hilbert space, $\psi : \mathcal{X} \to H$ a feature map, and $K(x, x') = \langle \psi(x), \psi(x') \rangle_H$ a kernel on $\mathcal{X}$. The RKHS $\mathcal{H}$ of $K$ consists of functions $f = \langle g, \psi(\cdot) \rangle_H$, with norm*

$$\|f\|_{\mathcal{H}} = \inf\{\|g'\|_H : g' \in H \text{ s.t. } f = \langle g', \psi(\cdot) \rangle_H\} \tag{16}$$

We also state here the generalization bound for kernel ridge regression used in Section 4, adapted from Bach (2021, Proposition 7.1).

**Proposition 7** (Generalization bound for kernel ridge regression). *Denote $f^*(x) = \mathbb{E}_\rho[y|x]$. Assume $f^* \in \mathcal{H}$, $\mathrm{Var}_\rho[y|x] \leq \sigma^2$, $K(x, x) \leq 1$ a.s., and define*

$$\hat{f}_\lambda := \arg\min_{f \in \mathcal{H}} \frac{1}{n} \sum_{i=1}^n (y_i - f(x_i))^2 + \lambda\|f\|_{\mathcal{H}}^2, \tag{17}$$

*for i.i.d. data $(x_i, y_i) \sim \rho$, $i = 1, \ldots, n$. Let*

$$n \geq \max\left\{ \frac{\|f^*\|_\infty}{\sigma^2}, \frac{\|f^*\|_{\mathcal{H}}^2}{\sigma^2 \mathbb{E}_{\rho_X}[K(x, x)]}, \frac{\sigma^2 \mathbb{E}_{\rho_X}[K(x, x)]}{\|f^*\|_{\mathcal{H}}^2} \right\}.$$

*For $\lambda = \sqrt{\sigma^2 \mathbb{E}_{\rho_X}[K(x, x)]/n\|f^*\|_{\mathcal{H}}^2}$, we have*

$$\mathbb{E}[R(\hat{f}_\lambda) - R(f^*)] \leq C\|f^*\|_{\mathcal{H}} \sqrt{\frac{\sigma^2 \mathbb{E}_{\rho_X}[K(x, x)]}{n}}, \tag{18}$$

*where $C$ is an absolute constant.*

*Proof.* Under the conditions of the theorem, we may apply (Bach, 2021, Proposition 7.1), which states that for $\lambda \leq 1$ and $n \geq \frac{5}{\lambda}(1 + \log(1/\lambda))$, we have

$$\mathbb{E}[R(\hat{f}_\lambda) - R(f^*)] \leq 16 \frac{\sigma^2}{n} \mathrm{Tr}((\Sigma + \lambda I)^{-1}\Sigma) + 16\lambda \langle f^*, (\Sigma + \lambda I)^{-1}\Sigma f^* \rangle_{\mathcal{H}} + \frac{24}{n^2}\|f^*\|_\infty^2,$$

where $\Sigma = \mathbb{E}_{\rho_X}[K(x, \cdot) \otimes K(x, \cdot)]$ is the covariance operator. We conclude by using the inequalities

$$\mathrm{Tr}((\Sigma + \lambda I)^{-1}\Sigma) \leq \frac{\mathrm{Tr}(\Sigma)}{\lambda} = \frac{\mathbb{E}_{\rho_X}[K(x, x)]}{\lambda}$$

$$\langle f^*, (\Sigma + \lambda I)^{-1}\Sigma f^* \rangle_{\mathcal{H}} \leq \|f^*\|_{\mathcal{H}}^2,$$

and optimizing for $\lambda$. $\square$

### E.1  PROOF OF PROPOSITION 1 (RKHS OF ONE-LAYER CONVOLUTIONAL KERNEL)

*Proof.* From Theorem 6, the RKHS contains functions of the form

$$f(x) = \langle F, A\Phi(x) \rangle_{L^2(\Omega_1, \mathcal{H})},$$

with RKHS norm equal to the minimum of $\|F\|_{L^2(\Omega_1, \mathcal{H})}$ over such decompositions.

We may alternatively write $f(x) = \langle G, \Phi(x) \rangle_{L^2(\Omega, \mathcal{H})}$ with $G = A^*F$. The mapping from $F$ to $G$ is one-to-one if $G \in \mathrm{Range}(A^*)$. Then, we obtain that equivalently, the RKHS contains functions of this form, with $G \in \mathrm{Range}(A^*)$, and with RKHS norm equal to the minimum of $\|A^{\dagger*}G\|_{L^2(\Omega_1, \mathcal{H})}$ over such decompositions. $\square$

### E.2 PROOF OF PROPOSITION 2 (RKHS OF 2-LAYER CKN WITH QUADRATIC $k_2$)

*Proof.* From Theorem 6, the RKHS contains functions of the form

$$f(x) = \langle F, A_2 M_2 P_2 A_1 \Phi_1(x) \rangle_{L^2(\Omega_2, \mathcal{H}_2)},$$

with RKHS norm equal to the minimum of $\|F\|_{L^2(\Omega_2, \mathcal{H}_2)}$ over such decompositions. Here, $\Phi_1(x) \in L^2(\Omega, \mathcal{H})$ is given by $\Phi_1(x)[u] = \varphi_1(x_u)$, so that $\Phi(x)$ in the statement is given by $\Phi(x) = \Phi_1(x) \otimes \Phi_1(x)$. We also have that $\mathcal{H}_2 = (\mathcal{H} \otimes \mathcal{H})^{|S_2| \times |S_2|}$, so that we may write $F = (F_{pq})_{p,q \in S_2}$ with $F_{pq} \in L^2(\Omega_2, \mathcal{H} \otimes \mathcal{H})$.

For $p, q \in S_2$, denoting by $L_c$ the translation operator $L_c x[u] = x[u - c]$, we have

$$(M_2 P_2 A_1 \Phi_1(x)[u])_{pq} = L_p A_1 \Phi_1(x)[u] \otimes L_q A_1 \Phi_1(x)[u]$$
$$= \mathrm{diag}(L_p A_1 \Phi_1(x) \otimes L_q A_1 \Phi_1(x))[u]$$
$$= \mathrm{diag}((L_p A_1 \otimes L_q A_1) \Phi(x))[u].$$

Then, we have

$$\langle F_{pq}, (A_2 M_2 P_2 A_1 \Phi_1(x))_{pq} \rangle_{L^2(\Omega_2, \mathcal{H} \otimes \mathcal{H})} = \langle F_{pq}, A_2 \mathrm{diag}((L_p A_1 \otimes L_q A_1) \Phi(x)) \rangle_{L^2(\Omega_2, \mathcal{H} \otimes \mathcal{H})}$$
$$= \langle A_2^* F_{pq}, \mathrm{diag}((L_p A_1 \otimes L_q A_1) \Phi(x)) \rangle_{L^2(\Omega_1, \mathcal{H} \otimes \mathcal{H})}$$
$$= \langle \mathrm{diag}(A_2^* F_{pq}), (L_p A_1 \otimes L_q A_1) \Phi(x) \rangle_{L^2(\Omega_1^2, \mathcal{H} \otimes \mathcal{H})}$$
$$= \langle (L_p A_1 \otimes L_q A_1)^* \mathrm{diag}(A_2^* F_{pq}), \Phi(x) \rangle_{L^2(\Omega^2, \mathcal{H} \otimes \mathcal{H})}.$$

We may then write this as $\langle G_{pq}, \Phi(x) \rangle_{L^2(\Omega^2, \mathcal{H} \otimes \mathcal{H})}$ with

$$G_{pq} = (L_p A_1 \otimes L_q A_1)^* \mathrm{diag}(A_2^* F_{pq}),$$

and the mapping between $F_{pq} \in L^2(\Omega_2, \mathcal{H} \otimes \mathcal{H})$ and $G_{pq} \in L^2(\Omega^2, \mathcal{H} \otimes \mathcal{H})$ is one-to-one if $G_{pq} \in \mathrm{Range}((L_p A_1 \otimes L_q A_1)^*)$, and $\mathrm{diag}((L_p A_1 \otimes L_q A_1)^{\dagger *} G_{pq}) \in \mathrm{Range}(A_2^*)$. We may then equivalently write the RKHS norm as the minimum over $G_{pq}$ satisfying such constraints for all $p, q \in S_2$, of the quantity

$$\sum_{p,q \in S_2} \|F_{pq}\|^2 = \sum_{p,q \in S_2} \|A_2^{\dagger *} \mathrm{diag}((L_p A_1 \otimes L_q A_1)^{\dagger *} G_{pq})\|_{L_2(\Omega_2, \mathcal{H} \otimes \mathcal{H})}^2.$$

$\square$

### E.3 PROOF OF PROPOSITION 5 (RKHS OF 3-LAYER CKN WITH QUADRATIC $k_{2/3}$)

*Proof.* Let $\Phi(x) = (\varphi_1(x_u))_u \in L^2(\Omega, \mathcal{H})$, so that we may write

$$\sum_{u_1, u_2, u_3, u_v \in \Omega} G[u_1, u_2, u_3, u_4](x_{u_1}, x_{u_2}, x_{u_3}, x_{u_4}) = \langle G, \Phi(x)^{\otimes 4} \rangle_{L^2(\Omega^4, \mathcal{H}^{\otimes 4})},$$

for some $G \in L^2(\Omega^4, \mathcal{H}^{\otimes 4})$.

From Theorem 6, the RKHS contains functions of the form

$$f(x) = \langle F, A_3 M_3 P_3 A_2 \Phi_2(x) \rangle_{L^2(\Omega_3, \mathcal{H}_3)}, \tag{19}$$

with RKHS norm equal to the minimum of $\|F\|_{L^2(\Omega_3, \mathcal{H}_3)}$ over such decompositions. Here, $\Phi_2(x) \in L^2(\Omega_1, \mathcal{H}_2) = L^2(\Omega_1, (\mathcal{H} \otimes \mathcal{H})^{|S_2| \times |S_2|})$ is given as in the proof of Proposition 2, by

$$\Phi_{2,qr}(x)[u] = \mathrm{diag}((L_q A_1 \otimes L_r A_1)(\Phi(x) \otimes \Phi(x)))[u],$$

for $q, r \in S_2$. A patch $P_3 A_2 \Phi_2(x)[u]$ is then given by

$$P_3 A_2 \Phi_2(x)[u] = (L_p A_2 \Phi_{2,qr}(x)[u])_{p \in S_3, q, r \in S_2} \in (\mathcal{H} \otimes \mathcal{H})^{|S_3| \times |S_2| \times |S_2|}.$$

Applying the quadratic feature map given by $\varphi_3(z) = z \otimes z \in (\mathcal{H}^{\otimes 4})^{(|S_3| \times |S_2| \times |S_2|)^2}$ for $z \in (\mathcal{H} \otimes \mathcal{H})^{|S_3| \times |S_2| \times |S_2|}$, we obtain for $\alpha = (p, q, r, p', q', r') \in (S_3 \times S_2 \times S_2)^2$,

$$(M_3 P_3 A_2 \Phi_2(x)[u])_\alpha = L_p A_2 \Phi_{2,qr}(x)[u] \otimes L_{p'} A_2 \Phi_{2,q'r'}(x)[u]$$
$$= \mathrm{diag}(A_{2,\alpha}(\Phi_{2,qr}(x) \otimes \Phi_{2,q'r'}(x)))[u],$$

where
$$A_{2,\alpha} = L_p A_2 \otimes L_{p'} A_2.$$
Now, one can check that we have the following relation:
$$\Phi_{2,qr}(x) \otimes \Phi_{2,q'r'}(x) = \mathrm{diag}((L_q A_1 \otimes L_r A_1)(\Phi(x) \otimes \Phi(x))) \otimes \mathrm{diag}((L_{q'} A_1 \otimes L_{r'} A_1)(\Phi(x) \otimes \Phi(x)))$$
$$= \mathrm{diag}_2(A_{1,\alpha}\Phi(x)^{\otimes 4}),$$
with
$$A_{1,\alpha} = L_q A_1 \otimes L_r A_1 \otimes L_{q'} A_1 \otimes L_{r'} A_1.$$

Since $\mathcal{H}_3 = (\mathcal{H}^{\otimes 4})^{(|S_3| \times |S_2| \times |S_2|)^2}$, we may write $F = (F_\alpha)_{\alpha \in (S_3 \times S_2 \times S_2)^2}$, with each $F_\alpha \in L^2(\Omega_3, \mathcal{H}^{\otimes 4})$. We then have
$$\langle F_\alpha, (A_3 M_3 P_3 A_2 \Phi_2(x))_\alpha \rangle_{L^2(\Omega_3)}$$
$$= \langle F_\alpha, A_3 \mathrm{diag}(A_{2,\alpha}\mathrm{diag}_2(A_{1,\alpha}\Phi(x)^{\otimes 4})) \rangle_{L^2(\Omega_3)}$$
$$= \langle \mathrm{diag}(A_3^* F_\alpha), A_{2,\alpha}\mathrm{diag}_2(A_{1,\alpha}\Phi(x)^{\otimes 4}) \rangle_{L^2(\Omega_2^2)}$$
$$= \langle \mathrm{diag}_2(A_{2,\alpha}^* \mathrm{diag}(A_3^* F_\alpha)), A_{1,\alpha}\Phi(x)^{\otimes 4} \rangle_{L^2(\Omega_1^4)}$$
$$= \langle A_{1,\alpha}^* \mathrm{diag}_2(A_{2,\alpha}^* \mathrm{diag}(A_3^* F_\alpha)), \Phi(x)^{\otimes 4} \rangle_{L^2(\Omega^4)}.$$
We may write this as $\langle G_\alpha, \Phi(x)^{\otimes 4} \rangle_{L^2(\Omega^4, \mathcal{H}^{\otimes 4})}$, with
$$G_\alpha = A_{1,\alpha}^* \mathrm{diag}_2(A_{2,\alpha}^* \mathrm{diag}(A_3^* F_\alpha)).$$
The mapping from $F_\alpha$ to $G_\alpha$ is bijective if $G_\alpha$ is constrained to the lie in the range of the operator $E_\alpha$. If $G_\alpha$ satisfies this constraint, we may write
$$F_\alpha = A_3^{\dagger *} \mathrm{diag}(A_{2,\alpha}^{\dagger *} \mathrm{diag}_2(A_{1,\alpha}^{\dagger *} G_\alpha)).$$
Then, the resulting penalty on $G_\alpha$ is as desired.

$\square$

### E.4 PROOF OF PROPOSITION 3 (GENERALIZATION FOR ONE-LAYER CKN)

*Proof.* Note that we have
$$\mathbb{E}_x[K_1(x,x)] = \sum_u \sum_{v,r} h[u-v]h[u-v-r]\,\mathbb{E}_x[k_1(x_v, x_{v-r})]$$
$$\leq \sum_{v,r} \langle h, L_r h \rangle \sigma_r^2$$
$$= |\Omega| \sum_r \langle h, L_r h \rangle \sigma_r^2.$$

It remains to verify that $\|f^*\|_{\mathcal{H}_{K_1}} = \sqrt{|\Omega|}\|g\|_{\mathcal{H}}$. Note that if we denote $G = (g)_{u \in \Omega} \in L^2(\Omega, \mathcal{H})$, then we have $A^*G = G$, since $\sum_v h[v-u]G[v] = (\sum_v h[v-u])g = g$. This implies that $A^{*\dagger}G = G$, regardless of which pooling filter is used. Then we have, by (5) that $\|f^*\|^2 \leq |\Omega|\|g\|^2$. Further, since $g$ is of minimal norm, no other $G \in L^2(\Omega, \mathcal{H})$ may lead to a smaller norm, so that we can conclude $\|f^*\|^2 = |\Omega|\|g\|^2$. $\square$

### E.5 PROOF OF PROPOSITION 4 (GENERALIZATION FOR TWO-LAYER CKN WITH QUADRATIC $k_2$)

*Proof.* We begin by studying the "variance" quantity $\mathbb{E}_x[K_2(x,x)]$. By expanding the construction of the kernel $K_2$, we may write
$$K_2(x,x) = \sum_{p,q \in S_2} \sum_{u,v,v'} h_2[u-v]h_2[u-v'] \times$$
$$\sum_{\substack{w_1, w_2 \\ w_1', w_2'}} h_1[v-w_1]h_1[v-w_2]h_1[v'-w_1']h_1[v'-w_2']k_1(x_{w_1-p}, x_{w_1'-p})k_1(x_{w_2-q}, x_{w_2'-q}).$$

Upper bounding the quantity $\mathbb{E}[k_1(x_{w_1-p}, x_{w_1'-p})k_1(x_{w_2-q}, x_{w_2'-q})]$ by 1 when $w_1 = w_1'$ and $w_2 = w_2'$, and by $\epsilon$ otherwise, the sum of the coefficients in front of 1 can be bounded as follows:

$$\sum_{p,q \in S_2} \sum_{u,v,v',w_1,w_2} h_2[u-v]h_2[u-v']h_1[v-w_1]h_1[v-w_2]h_1[v'-w_1]h_1[v'-w_2]$$

$$= |S_2|^2 \sum_{u,v,v'} \sum_v h_2[u-v]h_2[u-v]\langle L_v h_1, L_{v'} h_1 \rangle^2$$

$$= |S_2|^2 \sum_{v,v'} \langle L_v h_2, L_{v'} h_2 \rangle \langle L_v h_1, L_{v'} h_1 \rangle^2$$

$$= |\Omega||S_2|^2 \sum_v \langle h_2, L_v h_2 \rangle \langle h_1, L_v h_1 \rangle^2.$$

while the sum of coefficients bounded by $\epsilon$ is upper bounded by $|\Omega||S_2|^2$. Overall, this yields

$$\mathbb{E}_x[K_2(x,x)] \leq |S_2|^2|\Omega| \left( \sum_v \langle h_2, L_v h_2 \rangle \langle h_1, L_v h_1 \rangle^2 + \epsilon \right). \tag{20}$$

The bounds obtained for the example function $f^*(x) = \sum_{u,v} g(x_u, x_v)$ rely on plugging in the values of the Dirac filter $h[u] = \delta(u = 0)$ or average pooling filters $h[u] = 1/|\Omega|$ in the expression of $\mathbb{E}_x[K_2(x,x)]$, and on computing the norm $\|f^*\|_{K_2}$ for different architectures using Eq. (7) in Proposition 2. Computing $\mathbb{E}_x[K_2(x,x)]$ is immediate using the expression above. Bounding $\|f^*\|_{K_2}$ is more involved, and requires finding appropriate decompositions of the form $f^*(x) = \sum_{p,q} \sum_{u,v} G_{pq}[u,v](x_u, x_v)$ in order to leverage Proposition 2:

- When using a Dirac filter at the first layer, we need $|S_2| = |\Omega|$ in order to capture log-range interaction terms, and we represent $f^*$ as a sum of $G_p$ that are non-zero and equal to $g$ only on the $p$-th diagonal, *i.e.*, $G_p[u,v] = g$ when $v = u + p$, and zero otherwise. We then verify $\sum_p \sum_{u,v} G_p[u,v](x_u, x_v) = \sum_p \sum_u g(x_u, x_{u+p}) = f^*(x)$. Then, the expression (7) on this decomposition yields $|S_2||\Omega|\|g\|_{\mathcal{H}\otimes\mathcal{H}}^2 = |\Omega|^2\|g\|_{\mathcal{H}\otimes\mathcal{H}}^2$, for any choice of pooling $h_2$ such that $\|h_2\|_1 = 1$ (using similar arguments to the proof of Proposition 3). This is then equal to the squared norm of $f^*$ due to the minimality of $\|g\|_{\mathcal{H}\otimes\mathcal{H}}$.

- When using average pooling at the first layer and $|S_2| = |\Omega|$, we may use a single term $G[u,v]$ with all entries equal to $g$, *i.e.*, a decomposition $f^*(x) = \sum_{u,v} G[u,v](x_u, x_v)$. Using (7), we obtain an upper bound $|\Omega|\|g\|^2$ on the squared norm. The same decomposition can be used when $|S_2| = 1$, leading to the same bound.

$\square$

# F    GENERALIZATION GAINS UNDER SPECIFIC DATA MODELS

In this section, we consider simple models of architectures and data distribution where we may quantify more precisely the improvements in sample complexity guarantees thanks to pooling.

We consider architectures with non-overlapping patches, and a data distribution where the patches are independent and uniformly distributed on the sphere $\mathbb{S}^{d-1}$ sphere in $d := p|S_1|$ dimensions. Further, we consider a dot-product kernel on patches of the form $k_1(z, z') = \kappa(\langle z, z' \rangle)$ for $z, z' \in \mathbb{S}^{d-1}$, with the common normalization $\kappa(1) = 1$.

In the construction of Section 2, using non-overlapping patches corresponds to taking a downsampling factor $s_1 = |S_1|$ at the first layer, and a corresponding pooling filter such that $h_1[u] = 0$ for $u \neq 0$ mod $|S_1|$. Alternatively, we may more simply denote patches by $x_u$ with $u \in \Omega$ by considering a modified signal with more channels ($x_u = x[u]$ of dimension $pe$ instead of $p$, where $e$ is the patch size) so that extracting patches of size 1 actually corresponds to a patch of size $e$ of the underlying signal. We then have that the patches $x_u$ in a signal $x$ are i.i.d., uniformly distributed on the sphere $\mathbb{S}^{d-1}$. We denote the uniform measure on $\mathbb{S}^{d-1}$ by $d\tau$.

We note that when patches are in high dimension, overlapping patches may become near-orthogonal, which could allow extensions of our arguments below to the case with overlap, yet this may require different tools similar to Mei et al. (2021). We leave these questions to future work.

## F.1 QUANTIFYING THE TRACE OF THE COVARIANCE OPERATOR $E[K(x,x)]$

In this section, we focus on the "variance" term $\mathbb{E}[K(x,x)]$ which is used in the generalization results of Section 4.

**One layer.** In the one-layer case, we clearly have $\sigma_0^2 := \mathbb{E}[\kappa(\langle x_u, x_u \rangle)] = \kappa(1) = 1$. For $u \neq v$, since patches $x_u$ and $x_v$ are independent and i.i.d., $\sigma_{u-v}^2$ is a constant independent of $u, v$, which may be computed by integration on the sphere as:

$$\sigma_{u-v}^2 = \mathbb{E}[\kappa(\langle x_u, x_v \rangle)] = \mathbb{E}_{x_u \sim \tau}[\mathbb{E}_{x_v \sim \tau}[\kappa(\langle x_u, x_v \rangle)|x_u]] = \frac{\omega_{d-2}}{\omega_{d-1}} \int_{-1}^1 \kappa(t)(1-t^2)^{\frac{d-3}{2}} dt, \quad (21)$$

where we used a standard change of variable $t = \langle x_u, x_v \rangle$ when integrating $x_v$ over $\mathbb{S}^{d-1}$ (see, e.g., Efthimiou & Frye, 2014), with $\omega_{p-1} = 2\pi^{p/2}/\Gamma(p/2)$ the surface measure of the sphere $\mathbb{S}^{p-1}$. Note that the integral in (21) corresponds to the constant component in the Legendre decomposition of $\kappa$, which is known for common kernels as consider in various works studying spectral properties of dot-product kernels (Bach, 2017a; Minh et al., 2006). For instance, for the exponential kernel (or Gaussian on the sphere) $\kappa(\langle x, y \rangle) = e^{-\frac{\|x-y\|^2}{2\sigma^2}} = e^{\frac{1}{\sigma^2}(\langle x,y \rangle - 1)}$, which is used in most of our experiments with $\sigma = 0.6$, we have (Minh et al., 2006, Theorem 2):

$$\frac{\omega_{d-2}}{\omega_{d-1}} \int_{-1}^1 \kappa(t)(1-t^2)^{\frac{d-3}{2}} dt = e^{-1/\sigma^2}(2\sigma^2)^{(d-2)/2} I_{d/2-1}(1/\sigma^2)\Gamma(d/2),$$

where $I$ denotes the modified Bessel function of the first kind. For arc-cosine kernels, it may be obtained by leveraging the random feature expansion of the kernel (Bach, 2017a). More generally, we also note that when the patch dimension $d$ is large, we have

$$\frac{\omega_{d-2}}{\omega_{d-1}} \int_{-1}^1 \kappa(t)(1-t^2)^{\frac{d-3}{2}} dt \to \kappa(0), \quad \text{as } d \to \infty,$$

since $\frac{\omega_{d-2}}{\omega_{d-1}}(1-t^2)^{\frac{d-3}{2}}$ is a probability density that converges weakly to a Dirac mass at 0. In particular, for the exponential kernel with $\sigma = 0.6$, we have $\kappa(0) = e^{-1/\sigma^2} \approx 0.06$. When learning a translation-invariant function, the bound in Prop. 3 then shows that global average pooling yields an improvement w.r.t. no pooling of order $|\Omega|/(1+0.06|\Omega|)$. Note that removing the constant component of $\kappa$, i.e., using the kernel $\frac{\kappa(u)-\kappa(0)}{\kappa(1)-\kappa(0)}$, may further improve this bound, leading to a denominator very close to 1 when $d$ is large, and hence an improvement in sample complexity of order $|\Omega|$. We also remark that the dependence on $\kappa(0)$ may be removed by using a finer generalization analysis beyond uniform convergence that leverages spectral properties of the kernel (see Section F.2).

**Two layers with quadratic $k_2$.** For the two-layer case, we may obtain expressions of $\mathbb{E}[k_1(x_u, x_{u'})k_1(x_v, x_{v'})]$ as above. Denote $\epsilon := \mathbb{E}_{z,z' \sim \tau}[k_1(z, z')]$ where $z, z'$ are independent, which is given in (21). We may have the following cases:

- If $u = u'$ and $v = v'$, we have, trivially, $\mathbb{E}[k_1(x_u, x_{u'})k_1(x_v, x_{v'})] = 1$.
- If $u = u'$ and $v \neq v'$, we have $\mathbb{E}[k_1(x_u, x_{u'})k_1(x_v, x_{v'})] = \mathbb{E}[k_1(x_v, x_{v'})] = \epsilon$, since $x_v$ and $x_{v'}$ are independent. The same holds if $u \neq u'$ and $v = v'$.
- If $|\{u, v, u', v'\}| = 4$, we have

$$\mathbb{E}[k_1(x_u, x_{u'})k_1(x_v, x_{v'})] = \mathbb{E}[k_1(x_u, x_{u'})]\,\mathbb{E}[k_1(x_v, x_{v'})] = \epsilon^2.$$

- If $u = v$ and $|\{u, u', v'\}| = 3$, then we have

$$\mathbb{E}[k_1(x_u, x_{u'})k_1(x_v, x_{v'})] = \mathbb{E}_{x_u}[\mathbb{E}_{x_{u'}}[k_1(x_u, x_{u'})|x_u]\,\mathbb{E}_{x_v}[k_1(x_u, x_{v'})|x_u]] = \epsilon^2,$$

by using $\mathbb{E}_{x_{u'}}[k_1(x_u, x_{u'})|x_u] = \mathbb{E}_{z,z' \sim \tau}[k_1(z, z')]$, which holds by rotational invariance.

- If $u = v'$ and $u' = v$, we have $\mathbb{E}[k_1(x_u, x_{u'})k_1(x_v, x_{v'})] = \mathbb{E}_{z,z' \sim \tau}[k_1(z, z')^2] =: \tilde{\epsilon}$. This takes the same form as (21), but with a different kernel function $\kappa^2$ instead of $\kappa$. Note that in the case of the Exponential kernel, $\kappa^2$ is also an exponential kernel with different bandwidth.

Overall, when $\epsilon$ and $\tilde{\epsilon}$ are small compared to 1, we can see that the quantity $\mathbb{E}[k_1(x_u, x_{u'})k_1(x_v, x_{v'})]$ is small compared to 1 unless $u = u'$ and $v = v'$, thus satisfying the assumptions in Prop. 4. As described above, we may obtain expressions of $\epsilon$ and $\tilde{\epsilon}$ in various cases, and in particular these vanish in high dimension when using a kernel with $\kappa(0) = 0$.

## F.2 FAST RATES

In this section, we derive spectral decompositions of 1-layer CKN architectures with non-overlapping patches under the product of spheres distribution described in the previous section. This allows us to derive fast rates that depend on the complexity of the target functions on patches, and shows similar improvement factors to those derived in Section F.1, without the $\kappa(0)$ term, which in fact turns out to only be due to a single eigenspace, namely constant functions. We note that our derivation extends (Favero et al., 2021) to the case of generic pooling filters, and considers a different data distribution.

Before studying the 1-layer case, we remark that while it may seem natural to extend such decompositions to the 2-layer case using tensor products of spherical harmonics, as done by Scetbon & Harchaoui (2020) in the case without pooling, it appears that pooling may make it more challenging to find an eigenbasis since subspaces consisting of tensor products of spherical harmonics with fixed total degree are no longer left stable by the kernel.[5] We thus leave such a study to future work.

We begin by considering the following Mercer decomposition of the patch kernel

$$k_1(z, z') = \kappa(\langle z, z' \rangle) = \sum_{k=0}^{\infty} \mu_k \sum_{j=1}^{N(d,k)} Y_{k,j}(z)Y_{k,j}(z'),$$

where $Y_{k,j}$ for $k \geq 0$ and $j = 1, \ldots, N(d, k)$ are spherical harmonic polynomials of degree $k$ forming an orthonormal basis of $L^2(d\tau)$. The $\mu_k$ here are Legendre/Gegenbauer coefficients of the function $\kappa$ (see, *e.g.*, Bach, 2017a; Smola et al., 2001).

Note that the 1-layer kernel may be written as

$$K_h(x, y) = \sum_{u,v \in \Omega} h \circledast \bar{h}[u - v]\kappa(\langle x_u, y_v \rangle),$$

where $\circledast$ denotes circular convolution and $\bar{h}[u] := h[-u]$. We denote by $\tilde{e}_w[u] = \exp(2i\pi wu/|\Omega|)$, $w = 0, \ldots, |\Omega| - 1$, the DFT basis vectors, and by $e_w = \tilde{e}_w/\sqrt{|\Omega|}$ the normalized DFT basis vectors, which satisfy $\langle e_w, e_w \rangle = \sum_u e_w[u]e_w^*[u] = 1$, where $z^*$ is the complex conjugate of $z$. Define the Fourier coefficients

$$\hat{h}[w] = \langle h, \tilde{e}_w \rangle = \sum_u h[u]e^{-2i\pi wu/|\Omega|},$$

and let $\lambda_w := \widehat{h \circledast \bar{h}}[w] = |\hat{h}[w]|^2$. Note that when the filter is normalized s.t. $\|h\|_1 = 1$, we have $\lambda_0 = \hat{h}[0] = 1$. We will also use the Parseval identity $\|h\|_2^2 = (\sum_w \lambda_w)/|\Omega|$. Using the inverse DFT, it holds

$$h \circledast \bar{h}[u - v] = \frac{1}{|\Omega|} \sum_{w=0}^{|\Omega|-1} \lambda_w \tilde{e}_w[u - u] = \frac{1}{|\Omega|} \sum_{w=0}^{|\Omega|-1} \lambda_w \tilde{e}_w[u]\tilde{e}_w^*[v] = \sum_{w=0}^{|\Omega|-1} \lambda_w e_w[u]e_w^*[v].$$

Then, we have

$$K_h(x, y) = \sum_{w=0}^{|\Omega|-1} \sum_{k \geq 0} \lambda_w \mu_k \sum_{j=1}^{N(d,k)} \left( \sum_u e_w[u]Y_{k,j}(x_u) \right) \left( \sum_u e_w^*[u]Y_{k,j}(y_u) \right).$$

---

[5]For instance, the term $k_1(x_w, y_u)k_1(x_w, y_v)$, which may only appear in the presence of pooling, maps the polynomial $Y_k(x_u)Y_k(x_v)$ of degree $2k$ to a polynomial $\mu_k^2 Y_k(x_w)^2$ which is not necessarily orthogonal to all spherical harmonics tensor products of degree smaller than 2k.

Note that when $k = 0$, we have $Y_{0,1}(x_u) = 1$ for all $u$, hence

$$\left(\sum_u e_w[u]Y_{0,1}(x_u)\right)\left(\sum_u e_w^*[u]Y_{0,1}(y_u)\right) = \left(\sum_u e_w[u]\right)\left(\sum_v e_w^*[v]\right) = |\Omega|\,\mathbb{1}\{w = 0\},$$

since $e_0 = |\Omega|^{-1/2}(1, \dots, 1)$ and $\sum_u e_w[u] = 0$ for $w > 0$. Then, we may write

$$K_h(x, y) = |\Omega|\lambda_0\mu_0\phi_0(x)\phi_0^*(y) + \sum_{w=0}^{|\Omega|-1}\sum_{k \geq 1}\lambda_w\mu_k\sum_{j=1}^{N(d,k)}\phi_{w,k,j}(x)\phi_{w,k,j}^*(y),$$

with $\phi_0(x) = 1$, $\phi_{w,k,j}(x) = \sum_u e_w[u]Y_{k,j}(x_u)$, and $\phi^*$ denotes the complex conjugate of $\phi$.

It is then easy to check that the $\phi_0$ and $\phi_{w,k,j}$ form an orthonormal basis of $L^2(d\tau^{\otimes|\Omega|})$. We thus have obtained a Mercer decomposition of the kernel $K_h$ w.r.t. the data distribution, so that its eigenvalues are also the eigenvalues of the covariance operator (Caponnetto & De Vito, 2007), and control generalization performance, typically through the *degrees of freedom*

$$\mathcal{N}(\lambda) = \mathrm{Tr}((\Sigma + \lambda I)^{-1}\Sigma) = \sum_{m \geq 0}\frac{\xi_m}{\lambda + \xi_m},$$

where $\Sigma$ is the covariance operator, and $(\xi_m)_m$ is the collection of its eigenvalues. In particular, if we have a decay $\xi_m \asymp m^{-\alpha}$ (with $\alpha < 1$), then we have $\mathcal{N}(\lambda) \leq O(\lambda^{-1/\alpha})$, which then leads to a fast rate of $n^{-\alpha/(\alpha+1)}$ on the excess risk when optimizing for $\lambda$ in kernel ridge regression (Bach, 2021; Caponnetto & De Vito, 2007). In our case, the degrees of freedom takes the form

$$\mathcal{N}_h(\lambda) = \frac{|\Omega|\lambda_0\mu_0}{\lambda + |\Omega|\lambda_0\mu_0} + \sum_{w=0}^{|\Omega|-1}\sum_{k \geq 1}N(d,k)\frac{\lambda_w\mu_k}{\lambda + \lambda_w\mu_k} \tag{22}$$

We make a few remarks:

- Given that the number of spatial frequencies $w$ is fixed, the asymptotic decay rate of eigenvalues associated to the $\phi_{w,k,j}$ (that is, $\lambda_w\mu_k$, each with multiplicity $N(d,k)$) is the same as that of the eigenvalues associated to $\phi_{w_0,k,j}$ for some fixed $w_0$, which in turn corresponds to the decay for the corresponding dot-product kernel on the sphere. For instance, if $\kappa$ is the arc-cosine kernel, we have $\xi_m \asymp m^{-\alpha}$ with $\alpha = \frac{d+2}{d-1}$, and more generally $\alpha = \frac{2s}{d-1}$ for a kernel resembling a Sobolev space with $s$ bounded derivatives. This then leads to a rate $n^{2s/(2s+(d-1))}$, which only depends on the dimension $d$ of patches, rather than the full dimension $d|\Omega|$. One may also add more general assumption on the smoothness of the localized components of $f^*$ (such as $g$ in Proposition 3) in order to get rates that depend explicitly on the order of smoothness $s$ of such components (as in Caponnetto & De Vito, 2007).

- The eigenvalue associated to $\phi_0$ plays a minor role as by itself as it only contributes at most $\tau_\rho^2/n$ to the excess risk, which is negligible compared to the rest of the eigenvalues which lead to a slower $n^{-\alpha/(\alpha+1)}$ rate.

- With no pooling ($h$ is a Dirac delta), we have $\lambda_w = |\hat{h}[w]|^2 = 1$ for all $w$. We then have

$$\mathcal{N}_h(\lambda) \leq 1 + |\Omega|\mathcal{N}_\kappa(\lambda),$$

where we defined $\mathcal{N}_\kappa(\lambda) := \sum_{k \geq 1}N(d,k)\mu_k/(\lambda + \mu_k)$.

- With global pooling ($h = 1/|\Omega|$ is constant), we have $\lambda_0 = 1$, and $\lambda_w = 0$ for $w > 0$. This yields

$$\mathcal{N}_h(\lambda) \leq 1 + \mathcal{N}_\kappa(\lambda). \tag{23}$$

This then yields an improvement by a factor $|\Omega|$ in sample complexity guarantees compared to the scenario above with no pooling, namely the dominant term in the excess risk bound will be $C(1/n)^{\alpha/(\alpha+1)}$ compared to $C(|\Omega|/n)^{\alpha/(\alpha+1)}$.

- For more general pooling, one may exploit specific decays of $\lambda_w$ to obtain finer bounds. We may also obtain the following bound by Jensen's inequality

$$\mathcal{N}_h(\lambda) \leq 1 + |\Omega| \sum_{k \geq 1} N(d,k) \frac{\bar{\lambda}\mu_k}{\lambda + \bar{\lambda}\mu_k}$$

$$\leq 1 + |\Omega| \mathcal{N}_\kappa \left( \frac{\lambda}{\|h\|_2^2} \right)$$

where we used that $\bar{\lambda} = (\sum_w \lambda_w)/|\Omega| = \|h\|_2^2$, by Parseval's identity. When $\mathcal{N}_\kappa(\lambda) \leq C_\kappa \lambda^{-1/\alpha}$, we get a bound

$$\mathcal{N}_h(\lambda) \leq 1 + C_\kappa |\Omega| \|h\|_2^{2/\alpha} \lambda^{-1/\alpha}. \tag{24}$$

instead of a bound $\mathcal{N}(\lambda) \leq 1 + C|\Omega|\lambda^{-1/\alpha}$ for the case of no pooling, that is, the improvement is again controlled by $\|h\|_2^2$, which goes from $1/|\Omega|$ for global pooling, to $1$ for no pooling.

The following result provides an example of a generalization bound for invariant target functions, which illustrates that there is no curse of dimensionality in the rate if the patch dimension is much smaller than the full dimension (*i.e.*, $d \ll d|\Omega|$), as well as the benefits of pooling.

**Theorem 8** (Fast rates for one-layer CKN on invariant targets). *Consider an invariant target of the form $f^*(x) = \sum_{u \in \Omega} g^*(x_u)$, with $\mathbb{E}_{z \sim d\tau}[g^*(z)] = 0$, and assume:*

- *(capacity condition) $\mathcal{N}_\kappa(\lambda) \leq C_\kappa \lambda^{-1/\alpha}$,*

- *(source condition) $g^* = T_\kappa^r g_0$ and $\|g_0\|_{L^2(d\tau)} \leq C_*$, where $T_\kappa$ is the integral operator of the kernel $\kappa$ on $L^2(d\tau)$, with $r > \frac{\alpha-1}{2\alpha}$.*

*Then, kernel ridge regression with the one-layer CKN kernel $K_h$ with pooling filter $h$ (with $\|h\|_1 = 1$) satisfies, for $n$ large enough,*

$$\mathbb{E}[R(\hat{f}_n)] - R(f^*) \leq C|\Omega| \left( \frac{\|h\|_2^{2/\alpha}}{n} \right)^{\frac{2\alpha r}{2\alpha r + 1}}, \tag{25}$$

*where $C$ is independent of $|\Omega|$ and $h$. For global pooling, the factor $\|h\|_2^{2/\alpha} = |\Omega|^{-1/\alpha}$ can be improved to $|\Omega|^{-1}$. In contrast, with no pooling we have $\|h\|_2^{2/\alpha} = 1$, i.e., $n$ needs to be $|\Omega|$ times larger for the same guarantee. Note that for $\alpha \to 1$ and $r = 1/2$, the resulting bound resembles that of Proposition 3.*

*We note that if $g^*$ is assumed to be $s$-smooth on the sphere, the source condition with $2\alpha r = \frac{2s}{d-1}$ corresponds to a Sobolev condition of order $s$, and leads to the bound*

$$\mathbb{E}[R(\hat{f}_n)] - R(f^*) \leq C|\Omega| \left( \frac{\|h\|_2^{2/\alpha}}{n} \right)^{\frac{2s}{2s+d-1}}, \tag{26}$$

*which highlights that the rate only depends on the patch dimension $d$ instead of the full dimension $d|\Omega|$.*

*Proof.* Under the conditions of the theorem, we may apply (Bach, 2021, Proposition 7.2), which states that for $\lambda \leq 1$ and $n \geq \frac{5}{\lambda}(1 + \log(1/\lambda))$, we have

$$\mathbb{E}[R(\hat{f}_\lambda)] - R(f^*) \leq 16\frac{\tau_\rho^2}{n}\mathcal{N}_h(\lambda) + 16A_h(\lambda, f^*) + \frac{24}{n^2}\|f^*\|_\infty^2, \tag{27}$$

where the degrees of freedom $\mathcal{N}_h(\lambda)$ is given in (22) and satisfies the upper bound (24), and the approximation error is given by

$$A_h(\lambda, f^*) = \min_{f \in \mathcal{H}_K} \|f - f^*\|_{L^2(d\tau^{\otimes|\Omega|})}^2 + \lambda\|f\|_{\mathcal{H}_{K_h}}^2,$$

where $\mathcal{H}_{K_h}$ is the RKHS of $K_h$. Denote by

$$f = a_0\phi_0 + \sum_{w,k,j} a_{w,k,j}\phi_{w,k,j}, \qquad f^* = a_0^*\phi_0 + \sum_{w,k,j} a_{w,k,j}^*\phi_{w,k,j}$$

the decompositions of $f$ and $f^*$ in the orthonormal basis defined above. If $g^* = \sum_{k,j} g_{k,j} Y_{k,j}$ is the spherical harmonic decomposition of $g^*$, then we have

$$a_0^* = \mathbb{E}[f^*(x)] = 0$$

$$a_{0,k,j}^* = \mathbb{E}[f^*(x)\phi_{0,k,j}^*(x)] = g_{k,j} \sum_u e_0^*[u] = \sqrt{|\Omega|} g_{k,j}$$

$$a_{w,k,j}^* = 0 \quad \text{for } w \neq 0.$$

This yields

$$
\begin{aligned}
A_h(\lambda, f^*) &= \min_{a_0, a_{0,k,j}} (a_0 - a_0^*)^2 + \lambda \frac{a_0^2}{|\Omega|\lambda_0\mu_0} + \sum_{k\geq 1} \sum_{j=1}^{N(d,k)} (a_{0,k,j} - a_{0,k,j}^*)^2 + \lambda \frac{a_{0,k,j}^2}{\lambda_0\mu_k} \\
&= \min_{b_{k,j}} \sum_{k\geq 1} \sum_{j=1}^{N(d,k)} |\Omega|(b_{k,j} - g_{k,j})^2 + \lambda|\Omega| \frac{b_{k,j}^2}{\lambda_0\mu_k} \\
&= |\Omega| \min_{b_{k,j}} \sum_{k\geq 0} \sum_{j=1}^{N(d,k)} (b_{k,j} - g_{k,j})^2 + \lambda \frac{b_{k,j}^2}{\mu_k} \\
&= |\Omega| \min_{g \in \mathcal{H}} \|g - g^*\|_{L^2(d\tau)}^2 + \lambda \|g\|_{\mathcal{H}}^2 = |\Omega| A_\kappa(\lambda, g^*)
\end{aligned}
$$

where the second line uses $a_0^* = 0$ and considers $b_{k,j} = a_{0,k,j}/\sqrt{|\Omega|}$, while the third line uses $g_{0,1} = 0$ and the fact that $\lambda_0 = 1$ regardless of the choice of pooling filter $h$. Thus, $A_h(\lambda, f^*)$ does not depend on $h$, and corresponds to the approximation error $A_\kappa(\lambda, g^*)$ of the patch kernel $\kappa$ on the sphere (up to a factor $|\Omega|$). Under the source condition, we then have, by (Cucker & Smale, 2002, Theorem 3, p.33),

$$A_h(\lambda, f^*) = |\Omega| A_\kappa(\lambda, g^*) \leq |\Omega| C_*^2 \lambda^{2r}.$$

with $a_{0,k,j}^* = a_0^*$. Combining with (24) and plugging this into (27), we obtain

$$\mathbb{E}[R(\hat{f}_\lambda)] - R(f^*) \lesssim \frac{\tau_\rho^2}{n} + \frac{\tau_\rho^2 C_\kappa |\Omega| \|h\|_2^{2/\alpha} \lambda^{-1/\alpha}}{n} + |\Omega| C_*^2 \lambda^{2r} + \frac{\|f^*\|_\infty^2}{n^2}.$$

For the choice

$$\lambda_n = \left( \frac{\tau_\rho^2 C_\kappa |\Omega| \|h\|_2^{2/\alpha}}{r\alpha|\Omega|C_*^2 n} \right)^{\frac{\alpha}{2\alpha r + 1}},$$

we have

$$
\begin{aligned}
\mathbb{E}[R(\hat{f}_\lambda)] - R(f^*) &\lesssim (|\Omega|C_*^2)^{\frac{1}{2\alpha r+1}} \left( \frac{C_\kappa \tau_\rho^2 |\Omega| \|h\|_2^{2/\alpha}}{n} \right)^{\frac{2\alpha r}{2\alpha r+1}} + \frac{\tau_\rho^2}{n} + \frac{\|f^*\|^2}{n^2} \\
&= (C_\kappa \tau_\rho^2)^{\frac{2\alpha r}{2\alpha r+1}} C_*^{\frac{2}{2\alpha r+1}} |\Omega| \left( \frac{\|h\|_2^{2/\alpha}}{n} \right)^{\frac{2\alpha r}{2\alpha r+1}} + \frac{\tau_\rho^2}{n} + \frac{\|f^*\|^2}{n^2}.
\end{aligned}
$$

In the case of global pooling, the term in parentheses scales as $(|\Omega|^{-1/\alpha}/n)^{\frac{2\alpha r}{2\alpha r+1}}$, but can be improved to $(1/|\Omega|n)^{\frac{2\alpha r}{2\alpha r+1}}$ by using the bound (23) on $\mathcal{N}_h(\lambda)$ instead of (24).

When $r > \frac{\alpha-1}{2\alpha}$, we can choose $n$ large enough so that $n \geq \frac{5}{\lambda_n}(1 + \log(1/\lambda_n))$ is satisfied, and the higher order terms in $n$ are negligible. $\qquad\square$

