# OpenReview forum: "Approximation and Learning with Deep Convolutional Models: a Kernel Perspective"
_ICLR.cc/2022/Conference — ICLR 2022 Poster_

### Official Review · Reviewer_GUNE · 2021-10-27

**Correctness:** 4
**Technical Novelty And Significance:** 3
**Empirical Novelty And Significance:** 3
**Recommendation:** 8
**Confidence:** 3

**Main Review:**

Caveat: the paper is slightly over my head. I made efforts to dig into the high level ideas and proof strategies, but I probably cannot judge many of the aspects the paper contains.

Strengths:

* The paper studies the important question of why, from a theoretical perspective, CNNs are a good inductive bias.
* The idea of using simple kernels and the efficient learning algorithms as a surrogate is very neat, as it allows to employ RKHS theory (characterizing the space, generalization bounds).
* Well written throughout.


Weak points:
* The paper is hard to read if not very familiar with the discussed topics. A few high level "so what" discussions after Prop1/2 might help less familiar readers.
* Little context is provided in Sec 2, maybe move some context from Appendix into main paper?
* Error bars in would be useful as many results are close to each other.

**Summary Of The Paper:**

The paper aims to study properties of deep convolutional models via the surrogate of simple hierarchical kernels with convolution and pooling layers.
The authors characterise the underlying RKHS and their norms, and provide generalization bounds.
The results imply that convolution operations such as pooling and patches, if present in the data, lead to improve guarantees.
An empirical study both justifies studying these simple kernels in the first place, and illustrates the obtained theoretical results.


**Summary Of The Review:**

The paper addresses an important question, is well motivated, and carried out very thoroughly.

---

> ### Author Response · Authors · 2021-11-23
> **Response to Reviewer GUNE**
>
> Thank you for the insightful review and positive comments.
>
> We respond to your last three points in order:
>
> - Thank you for your suggestion. We added brief "so what" descriptions of Prop 1/2 just before their statements.
> - Thank you for suggesting to add more context in Section 2. We agree this would be useful, though the space limitations make this somewhat challenging. We included some brief additional comments on the motivations in the beginning of the section, and will consider re-organizing parts of the paper to improve this aspect in a final version.
> - error bars: thank you for this suggestion. We will add error bars for the results on 10k datapoints in the final version, by averaging over different choices of training sets.

---

### Official Review · Reviewer_FaP6 · 2021-10-31

**Correctness:** 3
**Technical Novelty And Significance:** 4
**Empirical Novelty And Significance:** 4
**Recommendation:** 6
**Confidence:** 4

**Main Review:**

Theoretical results on the approximation and generalization of convolutional kernels are limited in the literature. The reason why the convolutional kernel works well for image dataset is an interesting question. This paper made some progress in this direction by analyzing the RKHS of the 1 and 2 layers convolutional kernels. Furthermore, the experimental results are also interesting: a shallow convolutional kernel shares the same performance of deep convolutional kernels on CIFAR10 dataset.


However, many notations in this paper are not stated clearly, and the presentation is not in a clean way, which makes the paper hard to read. For example, I feel that Proposition 4 would be a very interesting result, but it is not very easy to understand. First, it is hard to find in the paper where h1 and h2 are defined, and what are their dimensions. Second, the proposition didn't state the relationship of rho_X with E[k_1(...)k_1(...)] (I can infer it from Proposition 3, but a Proposition should be more or less self-contained). Finally, it is not very easy to understand the proof of Proposition 4, especially the second part.

Some other confusing notations:
1. In Proposition 1, should f(x) = <G, Phi(x)> = \sum_{u \in \Omega} G[u] \phi(x_u) ? (I have similar question in proposition 2) I don't understand the meaning of notation \sum_u G[u](x_u). What is the nature of G? Is it a matrix (what is the dimension) or a function (from where to where)?
2. Page 5, below the line "Two-layers with a quadratic kernel.", there is a Phi which should be Psi?


Overall, I feel this paper is very interesting. However, I don't feel that this paper is written clearly. I will only recommend a weak acceptance.



**Summary Of The Paper:**

This paper analyzed convolutional kernels. Experimentally, the authors showed that convolutional kernel formed by shallow convolutional neural networks (2 - 3 layers) performs as well as the state of art deep convolutional kernels (e.g., Shankar et. al.). The authors also provide an exact description of the RKHS functions and their norm, of the one layer convolutional kernel and two layer convolutional kernel with low degree polynomial activation function on the top layer. Finally, assuming that the target function has a specific form, then theoretical results can give the generalization upper bound of kernel ridge regression in terms of the sample complexity; this shows that using a proper architecture can same the sample complexity by a polynomial factor of the input size.


**Summary Of The Review:**

The results are interesting, but not clearly written.

---

> ### Author Response · Authors · 2021-11-23
> **Response to Reviewer FaP6**
>
> Thank you for your detailed review and useful comments.
>
> Thanks to your comments and those of other reviewers, we have (hopefully!) improved clarity in the updated submission. Let us know if there are any other parts of the paper which should be improved.
>
> - $\rho_X$: thank you for pointing this out, we apologize for the inconsistent notation. We have now added the sampling $x \sim \rho_X$ in expectations throughout Section 4.
> - proof of Proposition 4: based on your feedback, we have included more details on the proof, particularly for computing the norm of $f^*$.
> - notation for G: here, $G$ and $\Phi(x)$ are both elements of $L^2(\Omega, \mathcal H)$, i.e. signals on $\Omega$ with values in $\mathcal H$.
> Their inner product satisfies  $\langle G, \Phi(x) \rangle_{L^2(\Omega, \mathcal H)} = \sum_u \langle G[u], \varphi(x_u) \rangle_{\mathcal H} = \sum_u G[u] (x_u)$,
> where the last equality follows from the reproducing property in kernel methods. We added some more details to emphasize the inner product, including a footnote in Section 2 that defines the space $L^2(\Omega, \mathcal H)$. Let us know if this remains unclear, we'd be happy to add further clarifications.
> - $\Phi$ → $\Psi$: fixed, thank you for pointing this out!

---

### Official Review · Reviewer_GwRv · 2021-10-31

**Correctness:** 4
**Technical Novelty And Significance:** 3
**Empirical Novelty And Significance:** 3
**Recommendation:** 8
**Confidence:** 3

**Main Review:**

Overall, I think the paper makes a valuable contribution towards understanding the ingredients necessary for good kernel performance for image classification. My comments are directed primarily towards improving the presentation of the results for clarity and readability. Comments are roughly in paper order.
- Nit: In the first paragraph of the introduction the authors claim that convolution and pooling operations are “known to be crucial.” This would seem to imply that these operations are necessary for good performance; however, empirical successes indicate only that they are sufficient (perhaps some other totally different architecture could also be effective).
- The last paragraph on the first page discusses NTK and NNGP kernels jointly, without drawing a distinction between the two. This may confuse some readers given that the kernels considered in the paper are purely NNGP kernels (unless I am mistaken here).
- In the bulleted list of contributions, there is some tension between the first two bullets regarding the benefits of depth. The first bullet shows that even a relatively shallow architecture can perform well, whereas the second argues the benefits of increasing depth. After reading the full paper these bullets both make sense (given the size of the pooling filters), but on a first read they seem at odds and might benefit from a brief explanation.
- The third bullet in the list of contributions mentions “certain invariances” which are then not specified until several pages later. It would be helpful to briefly describe the invariances (e.g. spatial smoothness) early on.
- In section 3, I appreciate the periodic interpretations of the results; even more of these high-level “what the result means and why it matters” descriptions would be great. For example, a discussion of the assumption that patches lie on the sphere and the regularization operator T would be helpful. Also, proposition 2 is particularly difficult to parse given the chain of definitions embedded in the proposition statement. Perhaps defining the notation before using it might be easier to read?
- Figure 2 is a bit confusing to parse. What are p and q in this illustration? What are the axes in the illustration? [I think these questions answer each other, but the figure shouldn’t leave me guessing.]
- The last paragraph in section 3 mentions Table 4, but based on the surrounding text I suspect this should reference Table 2 instead.
- The experimental setup in section 5 specifies a one-versus-all classification approach with an exponential kernel with sigma=0.6. How were these design choices made?
- Figure 3: why is this only considering one class? Also, please add x-axis labels and a “takeaway” sentence to the caption. In the main text Figure 3 (left) is described as “learning curves”, but the x-axis is the number of training examples (usually I’ve seen learning curves refer to a plot of MSE vs training time, which wouldn’t make sense for a kernel method.)
- There are a few typos, e.g. “a a one-versus-all approach” in the experimental setup paragraph on page 8, “comparable comparable decays for various depths” in the role of pooling paragraph on page 9, and “patches to captures interactions” at the bottom of page 15.


**Summary Of The Paper:**

The paper studies the RKHS and generalization properties of convolutional kernel networks, the NNGP corresponding to CNNs and the class of kernels that achieve state of the art performance on image classification. Among the theoretical contributions are analysis of the regularization induced by pooling, and interactions between patches captured with iterated convolutions. The paper also includes experimental results on CIFAR10 matching the prior state of the art for a kernel method while using a shallower architecture, as well as some ablations on the size of the convolutional kernels, the size of the pooling filters, and the number of layers.


**Summary Of The Review:**

The paper makes a valuable contribution to our understanding of convolutional networks for image classification, via expanding the theoretical characterization and generalization guarantees of their associated GP kernels, and offering experiments with simple architectures that match the state of the art for a kernel method on CIFAR10. In the main review I offer some suggestions for improving the clarity of the presentation.

---

> ### Author Response · Authors · 2021-11-23
> **Response to Reviewer GwRv**
>
> Thank you for your careful review and positive comments.
>
> Here are our replies to your points in the same order:
>
> - "crucial": thank you for this remark, we rephrased this sentence accordingly.
> - NNGP: indeed, our description in Section 2 closely matches NNGP kernels, and we have now emphasized this connection more explicitly in the last paragraph of Section 2. Nevertheless, convolutional NTKs for a similar architecture (in terms of pooling and patches) would also involve related quantities, and we expect that when truncating the higher layer kernels to fixed degree polynomials (or alternatively, when considering polynomial activations), the functional spaces would be quite similar to what we study.
> - depth: by "benefits of depth" we mostly mean benefits of at least two (hidden) layers, compared to a single one (note that the term "depth" often refers to merely two hidden layers in studies of approximation benefits, e.g. [https://arxiv.org/abs/1512.03965](https://arxiv.org/abs/1512.03965)). We agree that this is somewhat confusing, and changed it with "benefits of multiple layers".
> - "certain invariances": thank you, we made this more precise.
> - section 3: thanks for this feedback. We restructured Prop. 2 making it hopefully more readable, and we clarified the role of the operator T. In particular, the assumption of patches on the sphere is not needed for homogeneous patch kernels (the spherical regularization operator simply acts on the restriction to the sphere, as explained in Appendix A.2)
> - Figure 2: thank you for pointing this out. We have clarified the caption, giving the precise values of p and q used for the plots.
> - Table 4 → Table 2: fixed, thank you!
> - experiment setup: the one-vs-all approach is similar to other works (e.g. Shankar et al., Li et al.), the choice sigma=0.6 seemed best in our initial experiments (we also tried 0.5, 0.55 and 0.65).
> - Figure 3: we changed the left figure to average MSE over 10 classes, which we agree is more natural than a single class (and leads to a very similar figure), and expanded the caption. To our knowledge, "learning curve" usually refers to error as a function of sample size.
> - Thank you for pointing out these typos - they are now fixed.

---

> > ### Comment · Reviewer_GwRv · 2021-11-28
> > **Response to authors**
> >
> > Thank you for updating the paper and responding to my comments. I have a few remaining comments on the updated Figure 3:
> > - The leftmost figure is labeled "average MSE" but it isn't clear what the average is over. I know from our discussion that this is the average over the 10 classes, but this should be stated clearly in the paper. Likewise, the x axis is "n" which I know means number of training examples but this might not be clear to future readers.
> > - I don't really understand what the middle figure is showing; is there some pattern between the architecture and the spectral decay? I don't see a clear pattern with respect to depth alone, and although there might be a pattern combining depth, patch size, and pooling size I don't think I could confidently detect it based on 4 curves. Can the authors include a clearer description of the "take-home message" from this figure, either in the caption or the main text (or both)?
> > - I believe that the rightmost figure is comparing Gaussian pooling to strided convolutions, which achieve downsampling without explicit pooling, and showing that Gaussian pooling produces a much steeper eigenvalue decay. If this interpretation is correct, it is slightly confusing that the caption refers to "strided pooling" (maybe this should say "strided convolutions"?).
> > - The axis labels are a bit small (this is a minor point)

---

> > > ### Author Response · Authors · 2021-11-29
> > > **Thank you**
> > >
> > > Thank you for your helpful comments! We will address them in the updated version.
> > >
> > > * We will improve the figure labels and will clarify the meaning of "average" in the caption.
> > > * As we mention in the paragraph "Role of pooling", this figure was mainly supposed to convey the fact that varying the depth and architecture does not have a large effect on the spectral decay as long as appropriate pooling is used, while removing pooling significantly slows down the decay. We agree that the current figure and caption are somewhat confusing in this respect, and will do our best to improve this in the final version, probably by merging curves from the middle and right figures into a single figure.
> > > * Thanks for pointing this out, we will change "strided pooling" to "no pooling" and mention that this corresponds to "strided convolutions" in the caption.
> > > * We will make the labels larger.
> > >
> > > Thanks again for helping us improve the clarity of the paper.

---

### Official Review · Reviewer_nNN4 · 2021-11-06

**Correctness:** 4
**Technical Novelty And Significance:** 4
**Empirical Novelty And Significance:** 4
**Recommendation:** 8
**Confidence:** 4

**Main Review:**

The paper is very well written, with detailed and interesting discussions, which makes it a very pleasant read. I really appreciated that 1) the kernel models studied in this paper are motivated by simulation, showing that they match state-of-the-art performance for kernel methods; and 2) ‘good practice’ (patch whitening, Gaussian filters…) does improve a lot the performance of kernel methods (here, by providing a kernel architecture much simpler than the state-of-the-art kernel), which motivates a theoretical analysis to understand the reason, and hopefully give some clues on the properties of image function classes.
Here are a few comments and questions:

The approximation results are somewhat difficult to understand, but a lot of discussion is provided to parse them. However, some of the decompositions (1D or 2D discrete Fourier transforms) would gain at being worked out in more details (e.g., writing the computations etc more explicitly). For example, I think the independent regularization for 1-layer CKN, from $T$ (RKHS regularization of the patch function) + from $A$ (spatial regularization from pooling) to be quite nice. Similarly for the 2-layers case. I think those are really good intuitions on the type of regularization expected from pooling + kernels.

Why are the kernel methods so computationally expensive? I understand that one need to invert a $n$ by $n$ matrix, but for $n \approx 50000$ this should still be reasonable. What is the expensive step? Computing the feature map of the exponential kernel on the first layer? In that case, would it not be better to choose a (non-linear) feature map that is easy to compute, instead of the kernel? In general, what approach could help improve the computation of these hierarchical convolutional kernels?

In the examples provided in this work, no architecture achieves better performance than the Myrtle kernel. Do you believe that kernel methods already hit the wall of what data independent methods can achieve? From this work, is there any clue on what to do to improve kernel methods? (This is more open-ended and do not require a response).

From the experiments of this paper, and the characterization of the RKHS, can we understand the interactions etc. that are important to capture in CIFAR10 to get good accuracy. How is it relying on spatial regularities etc?


**Summary Of The Paper:**

In this paper, the authors study some simple convolutional kernels. They show that two or three layers convolutional kernels with polynomial kernels on the higher level and Gaussian pooling provide similar performance as the much more complicated state-of-the-art convolutional kernels (e.g., Myrtle kernel). Motivated by these good performance, they proceed to characterize the RKHS of these kernels, and describe how extra layers and pooling allows to capture interaction between patches with more or less spatial dependency. They then use the RKHS norm and some standard bound on the generalization error to show how choosing an architecture adapted to the target function can improve the statistical efficiency.

**Summary Of The Review:**

I think it is a very thorough work, with both lengthy discussions, experiments and theoretical analysis. The question of what accuracy can be achieved by kernel methods is important in its own right, and can potentially inform us on the performance of other methods (CNNs). Also we can hope that understanding theoretically the RKHS of good performing kernels can give us some clue on the properties of the image function classes. For these reasons, I recommend the paper to be accepted.

---

> ### Author Response · Authors · 2021-11-23
> **Response to Reviewer nNN4**
>
> Thank you for your thorough review and positive comments.
>
> We respond to your questions below:
>
> - computational cost: indeed, inverting the matrix is quite fast (a few minutes on a machine with large enough memory) — the expensive part is computing the elements of the kernel matrix, $K(x_i, x_j)$ for all pairs of images $(x_i, x_j)$. Each evaluation requires manipulating 4D tensors of initial size 32x32x32x32, and each patch extraction, non-linearity, and pooling operation requires non-trivial computations that were highly optimized in our code, but still are much more expensive than, say, evaluating a Gaussian kernel. Note that kernel approximations such as the Nystrom approach of Mairal (2016) make everything much cheaper to compute, but obtain lower accuracy (by 2-4%). It may also be possible to further optimize certain operations for the exact kernel, and to better leverage batching across different images, something which would be an interesting future direction.
> - performance of kernels: indeed, our improvement over Myrtle kernels is very minor (88.3% vs 88.2%), and leads us to believe that it may be hard to significantly improve the performance of kernels on Cifar10, at least using similar convolutional architectures. Perhaps further gains may be obtained with better pre-processing, or different kernel architectures that use other components beyond pooling and patches, and that may be tuned more specifically to the dataset.
> - Cifar10 performance: the question of using our characterization of the RKHS to better understand and interpret learned models on a given dataset is indeed a very interesting one. This seems difficult in the context of our experiments since the models we trained are using the representer theorem to represent the prediction functions as $f(x) = \sum_i \alpha_i K(x_i, x)$, which treats the kernel as a black-box and does not reveal much of the underlying structure. Studying this more carefully is nevertheless an interesting avenue for future work.

---

### Decision · Program_Chairs · 2022-01-20

**Decision:**

Accept (Poster)

**Comment:**

The paper addresses hierarchical kernels and provides an analysis of their RKHS along with generalization bounds and cases where improved generalization can be obtained. The reviewers appreciated the analysis and its implications. There were multiple concerns regarding presentation clarity, which the authors should address in the camera ready version.